# *Staphylococcus aureus* lipoproteins promote abscess formation in mice, shielding bacteria from immune killing

Majd Mohammad [1✉], Manli Na[1], Zhicheng Hu [1,2], Minh-Thu Nguyen [3,6], Pradeep Kumar Kopparapu[1], Anders Jarneborn[1,4], Anna Karlsson[1], Abukar Ali[1], Rille Pullerits [1,5], Friedrich Götz [3] & Tao Jin [1,4]

Despite being a major bacterial factor in alerting the human immune system, the role of *Staphylococcus aureus* (*S. aureus*) lipoproteins (Lpp) in skin infections remains largely unknown. Here, we demonstrated that subcutaneous injection of *S. aureus* Lpp led to infiltration of neutrophils and monocytes/macrophages and induced skin lesions in mice. Lipid-moiety of *S. aureus* Lpp and host TLR2 was responsible for such effect. Lpp-deficient *S. aureus* strains exhibited smaller lesion size and reduced bacterial loads than their parental strains; the altered phenotype in bacterial loads was TLR2-independent. Lpp expression in skin infections contributed to imbalanced local hemostasis toward hypercoagulable state. Depletion of leukocytes or fibrinogen abrogated the effects induced by Lpp in terms of skin lesions and bacterial burden. Our data suggest that *S. aureus* Lpp induce skin inflammation and promote abscess formation that protects bacteria from innate immune killing. This suggests an intriguing bacterial immune evasion mechanism.

[1] Department of Rheumatology and Inflammation Research, Institute of Medicine, Sahlgrenska Academy, University of Gothenburg, Gothenburg, Sweden. [2] Department of Microbiology and Immunology, The Affiliated Hospital of Guizhou Medical University, Guiyang, China. [3] Department of Microbial Genetics, Interfaculty Institute of Microbiology and Infection Medicine Tübingen (IMIT), University of Tübingen, Tübingen, Germany. [4] Department of Rheumatology, Sahlgrenska University Hospital, Gothenburg, Sweden. [5] Department of Clinical Immunology and Transfusion Medicine, Sahlgrenska University Hospital, Gothenburg, Sweden. [6] Present address: Section of Medical and Geographical Infectiology, Institute of Medical Microbiology, University Hospital of Münster, Münster, Germany. ✉email: majd.mohammad@rheuma.gu.se

Staphylococcus aureus (S. aureus), due to its ability to express many virulence factors and evade the human immune system, is a highly pathogenic bacterium, that can cause a broad spectrum of diseases in humans[1]. It is one of the most potent skin pathogens responsible for the majority of skin and soft tissue infections[2,3]. Apart from causing skin infections in chronic wounds[4], staphylococci also present major clinical challenges due to their postoperative infectious complications[5]. Of note, the pace of infections caused by S. aureus is rapidly increasing and the emergence of antibiotic-resistant bacterial strains, such as methicillin-resistant S. aureus (MRSA) has severely reduced the available treatment options[3,6].

During staphylococcal skin infection, microbe-associated molecular patterns (MAMPs) of S. aureus are recognized by pattern recognition receptors (PRRs), leading to activation of the host defense system[7]. As a result, a local immune reaction with the production of proinflammatory cytokines is initiated and recruitment of phagocytes to the local infection site is induced, thus giving rise to the local formation of skin abscesses or skin necrosis[8,9]. It is well-known that leukocytes, specifically neutrophils, are essential for the host defense to promote the formation of abscesses during S. aureus skin infections, a mechanism that is required to isolate the spread of the infection and ultimately enhance the bacterial eradication[3]. S. aureus itself also contributes to the abscess formation by producing specific virulence factors, which also play a role in structuring fibrin capsule that surround the abscess[10]. As a consequence, the pathogen can establish and shield itself from the host immune response and improve its own survival[11].

S. aureus expresses multiple virulence factors and bacterial molecules, including bacterial lipoproteins (Lpps), which are membrane anchored and represent a major class of surface proteins in S. aureus[12]. To date, up to 70 Lpps have been detected in S. aureus, and the number of Lpps varies within various S. aureus genomes[13]. In S. aureus, Lpps are tri-acylated and characterized by a long-chain fatty acid at the N-acyl position of the lipid moiety[14,15]. The long-chain N-acylated Lpp, recognized by TLR2-TLR1 receptors, silence both innate and adaptive immune responses[15]. Maturation of Lpp is known to be crucial for pathogenicity, inflammation, and immune signaling[12,16,17]. In addition, Lpps are known to have an essential role in nutrient and ion acquisition, thus playing a key role for the survival of the bacterium under infectious conditions[12,18]. Deficiency of pre-Lpp lipidation in various S. aureus mutant strains (Δlgt mutant) has diminished virulence compared to their parental strains due to reduced pathogenicity[12,18–20]. Recently, the role of S. aureus Lpp in various infection models, such as sepsis and infectious arthritis, have been the focus of many studies[18,20,21].

In the present study, we hypothesized that S. aureus Lpp could give rise to local skin inflammation, in a similar way to naturally occurring bacterial skin infections. Indeed, our results in a murine model demonstrate that subcutaneous injection of purified S. aureus Lpp induced skin inflammation via TLR2 with rapid infiltration of leukocytes to skin tissue, which led to an imbalance of local hemostasis toward a procoagulant state. Expression of Lpp in S. aureus increased the severity of skin infections and bacterial load in local tissues. Finally, we showed that depletion of leukocytes and fibrinogen abrogated the described effects induced by Lpp expression.

## Results

### S. aureus Lpp induce skin inflammation in a lipid-moiety dependent manner through TLR2.

To determine the effect of S. aureus Lpp in mice skin, the animals were subcutaneously (s.c.) injected with the purified lipoprotein, Lpl1, at concentrations of 1 μg, 5 μg or 20 μg per site and the size of skin lesion was assessed thereafter. Lpl1 is a model Lpp derived from the vSaα-specific lipoprotein-like cluster (lpl) that exists in highly pathogenic and epidemic S. aureus strains[17]. Within the first two days, the highest concentrations of Lpl1 (5 μg and 20 μg) induced skin lesion, whereas only mild and transient skin damage was observed in the mice that received the lowest dose (1 μg) of Lpl1 from day 3 postinjection (Fig. 1A). The skin lesions were significantly more pronounced from day 5 until termination day in the mice that received 20 μg of Lpl1 than those receiving 5 μg or 1 μg (Fig. 1A). The skin lesions increased with time in the group challenged with 20 μg, while the lesions successfully decreased in the groups injected with 5 μg and 1 μg, respectively (Fig. 1A, B).

To further understand whether the lipid- or protein-moiety of Lpp was responsible for the observed effect, Lpp lacking the lipid-moiety, Lpl1(−sp), or the intact Lpp, Lpl1(+sp), were injected s.c. into the mouse skin. Lpl1(−sp) completely lacked the capacity to induce abscess formation (Fig. 1C, D), suggesting that the lipid-moiety of S. aureus Lpp is fully responsible for inducing skin abscesses. These results show that purified S. aureus Lpp induce persistent skin inflammation, an effect mediated by their lipid content.

To assess the local inflammatory response in the skin tissue, we measured the levels of various chemokines in the supernatants of skin homogenates from mice s.c. injected with Lpl1(+sp), Lpl1 (−sp), and from healthy controls. Intriguingly, S. aureus Lpl1 (+sp) enhanced the levels of macrophage inflammatory protein-2 (MIP-2), keratinocyte chemoattractant (KC), and monocyte chemoattractant protein 1 (MCP-1) in a dose- and lipid-moiety dependent manner (Fig. 1E–G). This indicates that S. aureus Lpp induces a strong local release of leukocyte chemoattractants in the tissue, which triggers an influx of innate immune cells to this local inflammation site.

Since bacterial Lpp are the predominant ligands for TLR2[22–24], we next examined whether the Lpp-induced skin inflammation is mediated through this receptor. Both TLR2 deficient (TLR2−/−) and C57BL/6 wild-type mice were s.c. injected with purified Lpl1 and observed up to 10 days. Indeed, the TLR2−/− mice hardly presented any skin lesion during the entire course (Fig. 1H), suggesting that TLR2 is essential for the host signaling following Lpl1 injection. Furthermore, the chemokine level of MIP-2, KC and MCP-1 in the TLR2−/− mice was also downregulated, suggesting that the chemoattractant release was dependent on TLR2 (Fig. 1I–K).

### Neutrophils and monocytes rapidly migrate to the local skin following subcutaneous injection with S. aureus Lpp.

To elucidate the cellular mechanism behind the Lpp-induced inflammatory response in the skin, we injected Lpl1 into the mouse auricle and examined the presence of various immune cells in the local skin tissue by flow cytometry analysis one day after Lpl1 injection (Fig. 2). Skin tissues from the Lpl1-injected ears of C57BL/6 wild-type mice presented significantly higher frequencies of neutrophils (CD45+CD11b+Ly6G+), infiltrating monocytes (CD45+CD11b+Ly6G-F4/80+Ly6C$^{high}$) and skin macrophages (CD45+CD11b+Ly6G-F4/80+Ly6C$^{intermediate}$) than those of phosphate-buffered saline (PBS)-injected mouse ears (Fig. 2A–D). Such differences were not observed in the TLR2−/− mice (Fig. 2B–D). No differences were observed between the Lpl1-injected- and PBS-injected ears in neither the C57BL/6 wild-type nor the TLR2−/− mice with regard to NK cells (CD45+CD11b+Ly6G+CD335+) and T cells (CD45+CD11b-CD3+) (Fig. 2E, F). The representative FACS dot plots are shown for C57BL/6 wild-type mice and TLR2−/− mice, respectively (Fig. 2A, Supplementary Fig. 1). These results demonstrate that S. aureus Lpp

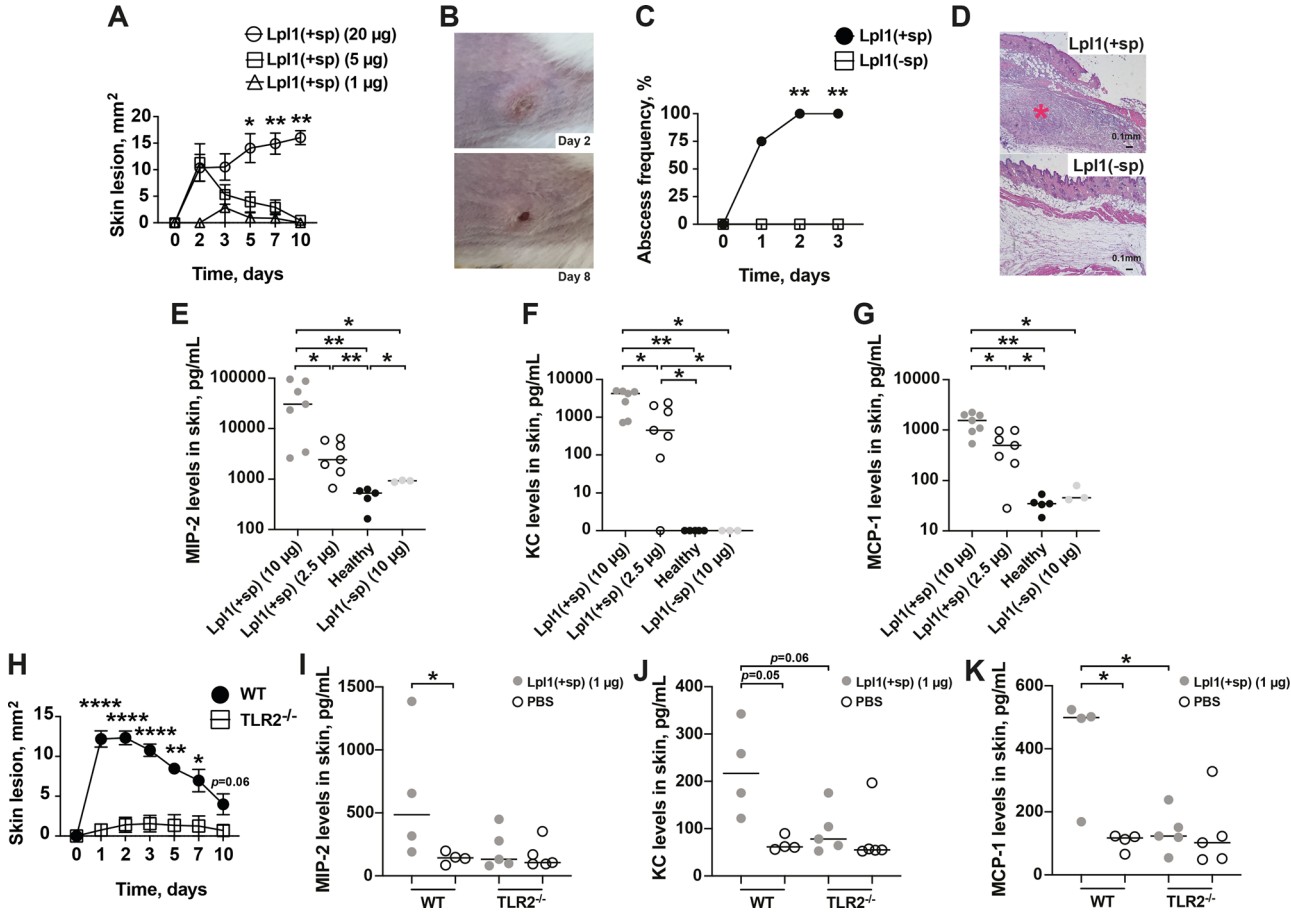

**Fig. 1 *S. aureus* Lpp induce skin inflammation in a lipid-moiety dependent manner through TLR2. A** The skin lesion size (mm$^2$) in NMRI mice ($n = 3$–12/group) up to 10 days after subcutaneous (s.c.) skin injection with 20 μl of purified *Staphylococcus aureus* lipoprotein, denoted as Lpl1(+sp), with a dose of 1 μg, 5 μg or 20 μg per site. **B** Representative images of mouse skin abscess formation on day 2 (upper panel), and mouse skin necrosis on day 8 (lower panel) after s.c. injection with 20 μg of purified Lpl1(+sp). **C** The frequency of abscess formation in NMRI mice ($n = 4$–5/group) following s.c. skin injection of Lpl1(+sp) or unlipidated Lpl1 protein, denoted as Lpl1(−sp) (10 μg/site). **D** Representative photomicrographs of mouse skin lesions on day 3 after s.c. injection with 10 μg of Lpl1(+sp) (upper panel) or Lpl1(−sp) (lower panel), stained with hematoxylin and eosin. Original magnification ×2.5. * Indicates inflamed skin tissue. The levels of **E** macrophage inflammatory protein-2 (MIP-2), **F** keratinocyte chemoattractant (KC), and **G** monocyte chemoattractant protein 1 (MCP-1) in the supernatant of skin biopsy homogenates on day 3 after s.c. skin injection with 20 μl of Lpl1(+sp) (2.5 or 10 μg; $n = 7$/group), Lpl1(−sp) (10 μg; $n = 3$) or healthy skin biopsy homogenates ($n = 5$) in NMRI mice. **H** The skin lesion size (mm$^2$) in C57BL/6 wild-type (WT) and TLR2 deficient (TLR2$^{-/-}$) mice up to 10 days after s.c. skin injection with 20 μl of Lpl1(+sp) (1 μg/site; $n = 10$/group). The levels of **I** MIP-2, **J** KC, and **K** MCP-1 in the supernatant of skin biopsy homogenates on day 3 after s.c. skin injection with 20 μl of Lpl1(+sp) (1 μg) or phosphate-buffered saline (PBS) ($n = 4$–5/group) in WT and TLR2$^{-/-}$ mice. The data were pooled from two independent experiments. Statistical evaluations were performed using the Fisher's exact test (**C**) or Mann–Whitney *U* test, with data expressed as the mean ± standard error of the mean (**A**, **H**), or presented as scatterplot with line indicating median value (**E**–**G**, **I**–**K**). *$P < 0.05$; **$P < 0.01$; ****$P < 0.0001$.

induce rapid influx of neutrophils and monocytes/macrophages upon exposure.

**S. aureus Lpps promote skin abscesses, prolong wound healing, and sustain higher bacterial burden in local skin.** In order to resemble and study the function of *S. aureus* Lpp in a real-life skin infection in mice, live *S. aureus* SA113 parental strain and SA113Δ*lgt* mutant strain lacking lipidated Lpp[16] were s.c. inoculated into the mouse skin. The SA113Δ*lgt* mutant strain led to significantly reduced skin lesions than those of the parental strain from day 3 up to day 10 postinfection (Fig. 3A). These results imply that *S. aureus* Lpp in live bacteria play an important role in skin infection and are in line with our previous finding with the purified *S. aureus* Lpp (Fig. 1). Also, the infection was resolved earlier in the SA113Δ*lgt* mutant group (Fig. 3B). The first lesion was found healed on day 11 and all skin lesions from the mice inoculated with SA113Δ*lgt* mutant strain were fully healed

by day 14 postinfection. In contrast, the first skin wound healing in mice inoculated with SA113 parental strain appeared on day 14, and only 50% of the skin wounds in this group were healed at the end of the experiment on day 21 postinfection (Fig. 3B).

To investigate whether the more severe skin lesions were associated with increased bacterial growth in the local skin tissue, the skin biopsies were collected on day 3 postinoculation and homogenized for colony-forming units (CFU) counts. In fact, the bacterial load was increased in the mice infected with the SA113 parental strain, suggesting that *S. aureus* Lpp provoked less bacterial eradication effect than its SA113Δ*lgt* mutant strain (Fig. 3C). Even with lower bacterial doses the SA113 parental strain gave rise to more severe skin lesions and tended to exhibit higher bacterial burden in contrast to its SA113Δ*lgt* mutant strain (Supplementary Fig. 2). Furthermore, the supernatant of the homogenized skin biopsies was used to assess the chemokine levels as well as to indirectly identify the amount of in situ

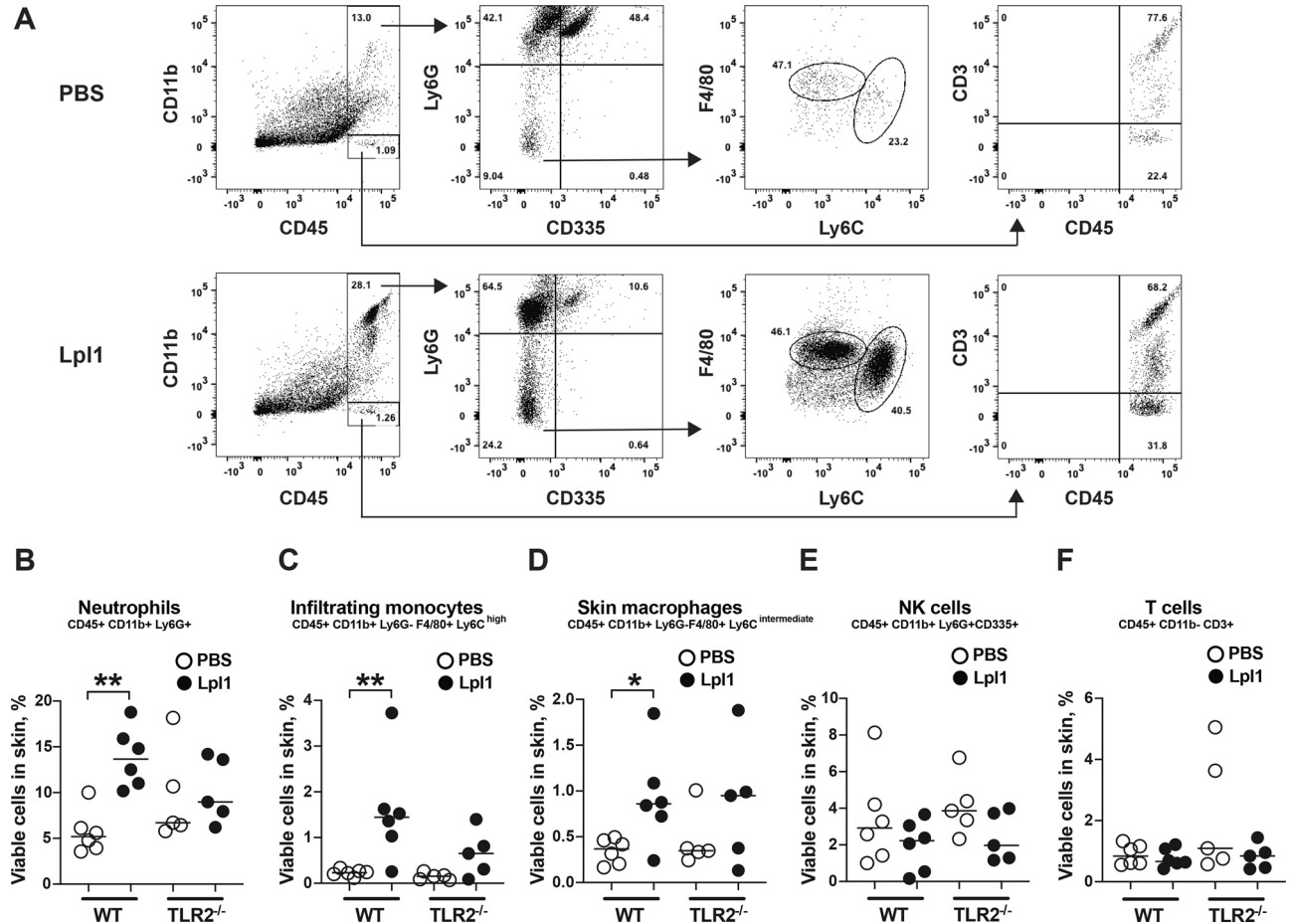

**Fig. 2 Neutrophils and monocytes rapidly migrate to the local skin following subcutaneous injection with *S. aureus* Lpp. A** Representative FACS gating strategy and dot plots of cells isolated from auricular skin tissue following subcutaneous (s.c.) injection of phosphate-buffered saline (PBS) (upper panel) or purified *Staphylococcus aureus* lipoprotein, denoted as Lpl1 (1 μg) (lower panel) in the mouse ear in C57BL/6 wild-type (WT) mice. Frequencies of infiltrating neutrophils (**B**), infiltrating monocytes (**C**), skin macrophages (**D**), NK cells (**E**) and T cells (**F**) in the auricular skin tissue of C57BL/6 WT and TLR2 deficient (TLR2$^{-/-}$) mice ($n = 5$–6/group) on day 1 after s.c. injection with 10 μl of PBS or Lpl1 (1 μg/site). The data were pooled from three independent experiments. Statistical evaluations were performed using the Mann–Whitney $U$ test, with data expressed as the median. *$P < 0.05$; **$P < 0.01$.

phagocytes by detecting the concentration of myeloperoxidase (MPO). The MIP-2 levels were downregulated in the SA113Δ*lgt* mutant strain compared to the SA113 parental strain (Fig. 3D). Although not significant, SA113Δ*lgt* mutant strain tended to induce lower levels of KC, MCP-1 and MPO in the skin tissue than the SA113 parental strain (Fig. 3E–G).

For complementation of the *lgt* mutant, a complemented mutant strain, SA113Δ*lgt* (pRB474::*lgt*), was then used to validate the impact of *S. aureus* Lpps in our infection model. Indeed, the mice inoculated with the complemented mutant strain exhibited more severe skin lesions as well as higher bacterial persistence in the local skin than those injected with the SA113Δ*lgt* mutant strain (Fig. 3H, I). The complemented mutant also led to higher levels of MIP-2, KC, MCP-1 as well as MPO in skin homogenates in contrast to the SA113Δ*lgt* mutant strain (Fig. 3J–M).

**S. aureus Lpp-induced fulminant bacterial growth is independent of host TLR2 signaling**. To study the importance of the Lpp–TLR2 interaction in the host model of live bacterial skin infection, TLR2$^{-/-}$ mice and C57BL/6 wild-type mice were s.c. inoculated with either Newman parental strain or NewmanΔ*lgt* mutant strain, which are *agr* positive *S. aureus* strains[25]. The wild-type mice infected with the NewmanΔ*lgt* mutant strain displayed smaller skin lesions than those infected with the Newman parental strain during the entire course of infection

(Fig. 4A), which is in agreement with the results observed using the *agr* negative *S. aureus* SA113 strain[25] (Fig. 3). Intriguingly, the skin lesions within the TLR2$^{-/-}$ mice inoculated with the Newman parental strain and NewmanΔ*lgt* mutant strain exhibited a similar lesion development pattern in the early course of disease up to day 5 postinfection. However, smaller skin lesions were observed in the NewmanΔ*lgt* mutant strain later in the disease course, on day 7 and 10 postinfection (Fig. 4B). These results suggest that *S. aureus* Lpp is important in the development of skin lesions mediated through TLR2 only at the early phase of infection. Moreover, the CFU counts from the local skin tissue in the Δ*lgt* mutant groups within both the C57BL/6 wild-type and TLR2$^{-/-}$ mice were lower than in those infected with the parental strain on day 3 and 10 postinfection (Fig. 4C, D). This demonstrates that *S. aureus* Lpps enhance the bacterial growth independently of host TLR2 signaling. Furthermore, the assessment of leukocyte chemoattractant release from the supernatant of skin biopsy homogenates on day 3 and 10 postinfection demonstrated that *S. aureus* Lpps triggered the release of MIP-2, KC and MCP-1 independently of TLR2 (Supplementary Fig. 3).

**Coinjection of Lpp and live S. aureus worsens the skin damage and increases the bacterial burden in local skin tissue**. To assess whether addition of exogenous Lpp to the Lpp-deficient strain gives rise to more severe skin damage and higher bacterial load,

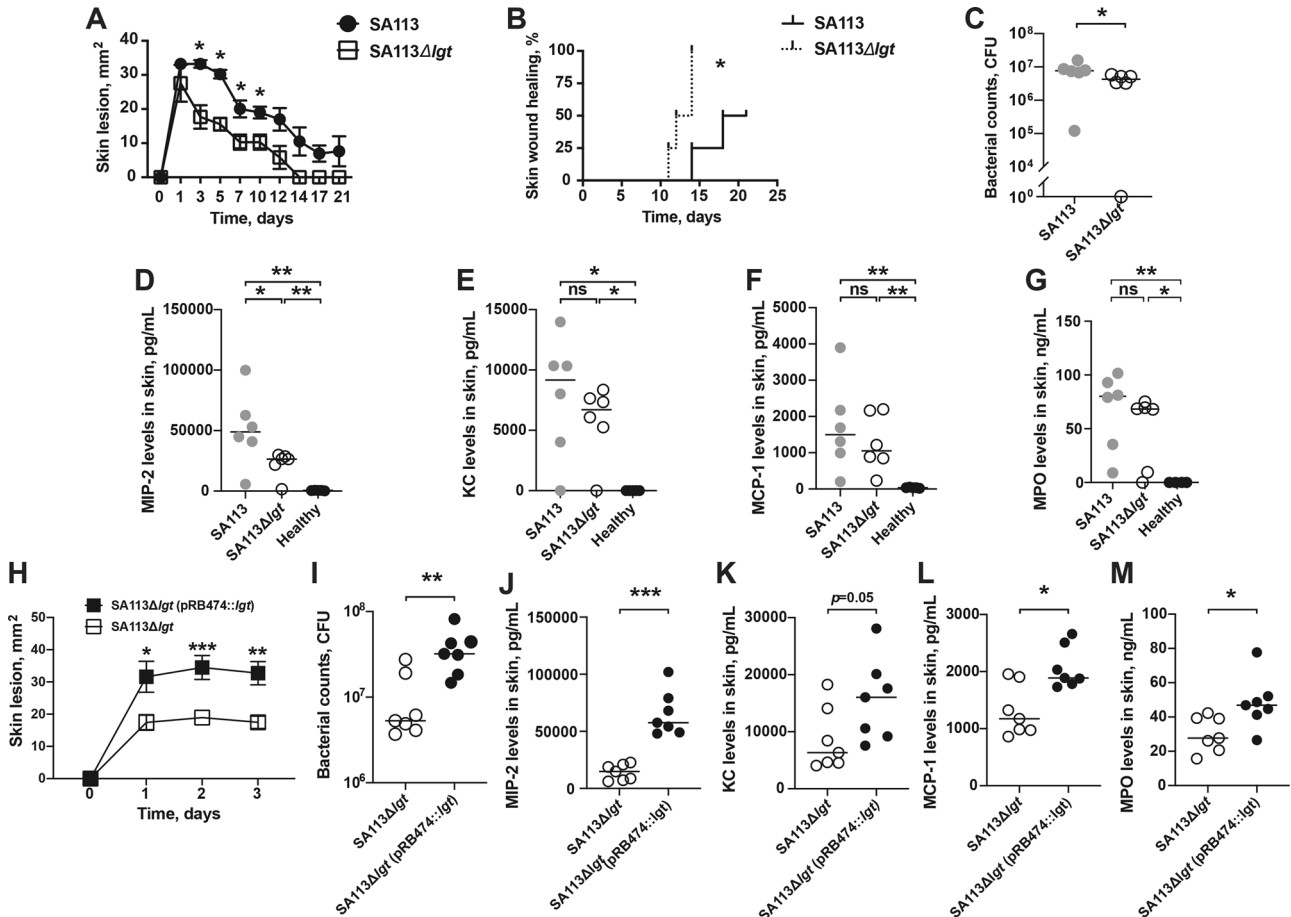

**Fig. 3 _S. aureus_ Lpp promote skin abscesses, prolong wound healing, and sustain higher bacterial burden in local skin.** The skin lesion size (mm$^2$) (**A**), and skin wound healing (**B**) in NMRI mice ($n = 4$/group) up to 21 days after subcutaneous (s.c.) skin injection with 50 µl of _Staphylococcus aureus_ SA113 parental strain or SA113Δ_lgt_ mutant strain ($7.5 \times 10^7$ colony-forming units [CFU]/site). Bacterial counts (**C**) and levels of **D** macrophage inflammatory protein-2 (MIP-2), **E** keratinocyte chemoattractant (KC), **F** monocyte chemoattractant protein 1 (MCP-1) and **G** myeloperoxidase (MPO) levels in the supernatant of skin biopsy homogenates from healthy NMRI mice ($n = 4$–5) or from mice on day 3 after s.c. skin infection with SA113 parental strain or SA113Δ_lgt_ mutant strain ($1.25 \times 10^8$ CFU/site; $n = 6$/group). The skin lesion size (mm$^2$) development in NMRI mice ($n = 7$/group) up to 3 days (**H**), and bacterial counts (**I**), levels of MIP-2 (**J**), KC (**K**), MCP-1 (**L**) and MPO (**M**) in the supernatant of skin biopsy homogenates on day 3 after s.c. skin infection with 50 µl of _Staphylococcus aureus_ SA113Δ_lgt_ mutant strain or complemented mutant SA113Δ_lgt_ (pRB474::_lgt_) strain ($7.5 \times 10^7$ CFU/site). The data were pooled from two independent experiments. Statistical evaluations were performed using the Mann–Whitney $U$ test, with data expressed as the mean ± standard error of the mean (**A**, **H**), or presented as scatterplot with line indicating median value (**C**–**G**, **I**–**M**). Statistical evaluations were performed using Mantel–Cox log-rank test (**B**). \*$P < 0.05$; \*\*$P < 0.01$; \*\*\*$P < 0.001$; ns = not significant.

live _S. aureus_ SA113Δ_lgt_ mutant strain and purified Lpl1(+sp) were s.c. coinjected into the mouse skin and the mice were followed for 3 days (Fig. 5). The mice coinjected with Lpl1(+sp) and SA113Δ_lgt_ mutant strain exhibited significantly bigger skin lesions during the entire course of infection than control mice coinjected with PBS and SA113Δ_lgt_ mutant strain (Fig. 5A). The frequency of skin abscess formation was higher on day 3 post-infection among the mice that received Lpl1(+sp) in combination with the live bacteria compared to the control group (80% vs 27%) (Fig. 5B). Moreover, the bacterial load in the local skin tissue was significantly higher in the mice that received the mixture containing purified Lpl1(+sp) (Fig. 5C).

To further justify that Lpp and fibrin capsules protect bacteria from killing by immune cells, skin biopsies were collected at an earlier time point (day 1 postinfection) when the fibrin capsules were not formed yet. The results demonstrated that there was no differences between the groups coinjected with Lpl1(+sp) and SA113Δ_lgt_ mutant strain compared to the control group coinjected with PBS and SA113Δ_lgt_ mutant strain (Supplementary Fig. 4).

Finally, the mice that were inoculated with a mixture of Lpl1 (+sp) and SA113Δ_lgt_ mutant strain presented enhanced MIP-2, KC, MCP-1, and MPO levels on day 3 postinfection compared to those that received a mixture of PBS and SA113Δ_lgt_ mutant strain (Fig. 5D–G), suggesting that addition of exogenous _S. aureus_ Lpp contributes to the release of large amounts of neutrophil- and monocyte chemoattractants, and to a higher degree of neutrophil infiltration in the local skin tissue.

To verify the essential role of the lipid-moiety of _S. aureus_ Lpps, mice were inoculated with a mixture of Lpl1(−sp) that only contains the protein-moiety, and SA113Δ_lgt_ mutant strain. No obvious differences were observed between the group coinjected with Lpl1(−sp) and SA113Δ_lgt_ mutant in contrast to the control group coinjected with PBS and SA113Δ_lgt_ mutant strain with regard to skin lesion size, bacterial count, and chemokine or MPO levels (Fig. 5H–M).

**Difference in bacterial clearance between the parental- and Δ_lgt_ mutant strain disappears by leukocyte depletion.** Since _S. aureus_ Lpp promoted recruitment of immune cells to the site of skin

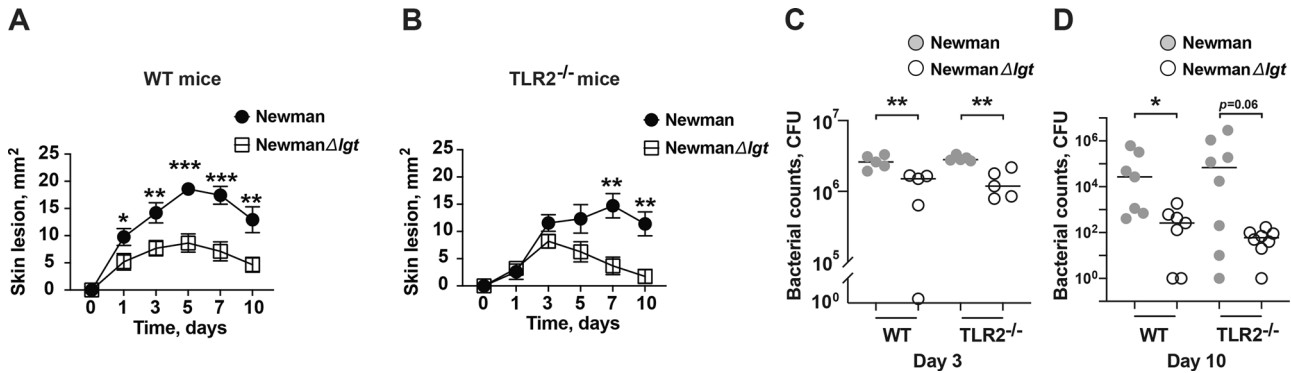

**Fig. 4 S. aureus Lpp-induced fulminant bacterial growth is independent of host TLR2 signaling.** The skin lesion size (mm²) in C57BL/6 wild-type (WT) mice (**A**) and TLR2 deficient (TLR2⁻/⁻) mice (**B**) up to 10 days after subcutaneous (s.c.) skin injection with 20 μl of *Staphylococcus aureus* Newman parental strain or NewmanΔ*lgt* mutant strain (1.5–2.0 × 10⁶ colony-forming units [CFU]/site; *n* = 13–14/group). Bacterial counts in the supernatant of skin biopsy homogenates on day 3 (*n* = 5/group) (**C**) and day 10 (*n* = 7–8/group) (**D**) after s.c. skin infection with Newman parental strain or NewmanΔ*lgt* mutant strain (1.5–2.0 × 10⁶ CFU/site). The data were pooled from two independent experiments. Statistical evaluations were performed using the Mann–Whitney *U* test, with data expressed as the mean ± standard error of the mean (**A, B**), or presented as scatterplot with line indicating median value (**C, D**). *P < 0.05; **P < 0.01; ***P < 0.001.

inflammation in the mice (Fig. 2), we next studied whether depletion of leukocytes influenced the clinical signs and bacterial burden in the local skin tissue. For that purpose, mice were pretreated with cyclophosphamide followed by s.c. infection with Newman parental strain or NewmanΔ*lgt* mutant strain. Indeed, leukocyte depletion in mice with cyclophosphamide treatment fully abrogated the *S. aureus* Lpp-induced effect in skin infection (Fig. 6). As expected, cyclophosphamide-treated mice developed more severe diseases. Sixty percent of mice died within the cyclophosphamide-treated group. The leukopenic mice also lost significantly more weight compared to the PBS-treated mice during the course of infection (Supplementary Fig. 5). While the two *S. aureus* strains formed similar skin lesions within the cyclophosphamide-treated mice, the Newman parental strain formed bigger lesions than those infected with the NewmanΔ*lgt* mutant strain in the PBS-treated mice during the course of infection (Fig. 6A). Interestingly, no significant difference was observed with regard to the bacterial load between the Newman parental strain and NewmanΔ*lgt* mutant strain in the leukocyte depleted cyclophosphamide-treated mice, whereas the bacterial counts were significantly higher in the PBS-treated mice infected with the parental strain compared to those infected with the Δ*lgt* mutant strain (Fig. 6B), demonstrating that the effect induced by Lpp expression is dependent on the presence of leukocytes.

Next, SCID mice, deficient in B- and T-cells, were used in order to study the role of the adaptive immunity on the phenotype that we have observed. More severe skin lesions (Fig. 6C) and higher bacterial load (Fig. 6D) was observed in the mice infected with the Newman parental strain than in those infected with the NewmanΔ*lgt* mutant strain in both the SCID mice as well as the Balb/c controls, indicating that the innate immunity rather than the adaptive immunity plays an important role in contributing to the effect induced by Lpp in skin infection.

**Lpp expression leads to an imbalanced coagulation/fibrinolysis hemostasis and fibrinogen depletion abrogates this Lpp-induced effect.** Considering the fact that more skin abscesses and bacterial loads were formed in the skin inoculated with the parental strains than those infected with the Δ*lgt* mutant strains (both SA113 and Newman strains), we hypothesized that *S. aureus* Lpp induced the formation of fibrin capsule in murine skin infection, thus protecting the bacteria from killing by immune cells. To provide evidence for our hypothesis, the levels of tissue factor (TF) and plasminogen activator inhibitor-1 (PAI-1) were

assessed in murine skin tissue on day 3 after s.c. injection of Lpl1. Intriguingly, Lpl1(+sp) dose-dependently enhanced the level of TF (Fig. 7A), and upregulated the expression of PAI-1 (Fig. 7B) in mice in a lipid-moiety dependent fashion. However, exogenous addition of Lpl1(+sp) to SA113Δ*lgt* mutant strain did not upregulate the expression of TF in local skin, as they did in PAI-1, as compared to their control groups (Fig. 7C, D). The levels of both TF (Fig. 7E) and PAI-1 (Fig. 7F) were fairly similar between the control group coinjected with PBS and SA113Δ*lgt* mutant strain as compared to the mixture containing Lpl1(-sp) and SA113Δ*lgt* mutant strain.

Since monocytes/macrophages are predominant producers of TF[26], we next investigated whether *S. aureus* Lpps induce the release of TF and PAI-1 in murine peritoneal macrophages. Unexpectedly, TF expression was not triggered by Lpl1 (Supplementary Fig. 6). On the other hand, the PAI-1 levels in the supernatant of peritoneal macrophages in wild-type mice were elevated upon stimulation of *S. aureus* Lpl1 and the synthetic lipopeptide, Pam3CSK4, whereas the corresponding stimuli led to similar PAI-1 levels as unstimulated ones in TLR2⁻/⁻ mice (Fig. 7G).

To further study the importance of hypercoagulation status in Lpp-induced effect, mice were treated with Ancrod in order to deplete fibrinogen and were thereafter s.c. infected with Newman parental strain or NewmanΔ*lgt* mutant strain. Strikingly, the differences between the skin lesions caused by parental strain and Δ*lgt* mutant strain were abolished during the course of infection in Ancrod-treated mice, while the differences remained consistent among the groups in the PBS-treated mice (Fig. 7H). Furthermore, the CFU counts in the local skin of mice inoculated with the Newman parental strain and NewmanΔ*lgt* mutant strain were similar within the fibrinogen-depleted mice while the bacterial burden was more pronounced in the mice infected with the Newman parental strain within the PBS-treated group (Fig. 7I). In order to assess whether bacterial dissemination occurred in fibrinogen-depleted mice upon s.c. infection with these *S. aureus* strains, the CFU counts in the blood, kidneys, liver, spleen, and lungs were performed. No CFU counts were found in the fibrinogen-depleted mice, while 1 out of 4 mice in the PBS-control group displayed positive counts in the kidneys, liver, spleen, and lungs (Supplementary Table 1). Figure 7J illustrates representative histological skin sections whereby the skin biopsies from mice infected with Newman parental strain displayed the most severe outcome with highly inflamed skin tissue surrounded

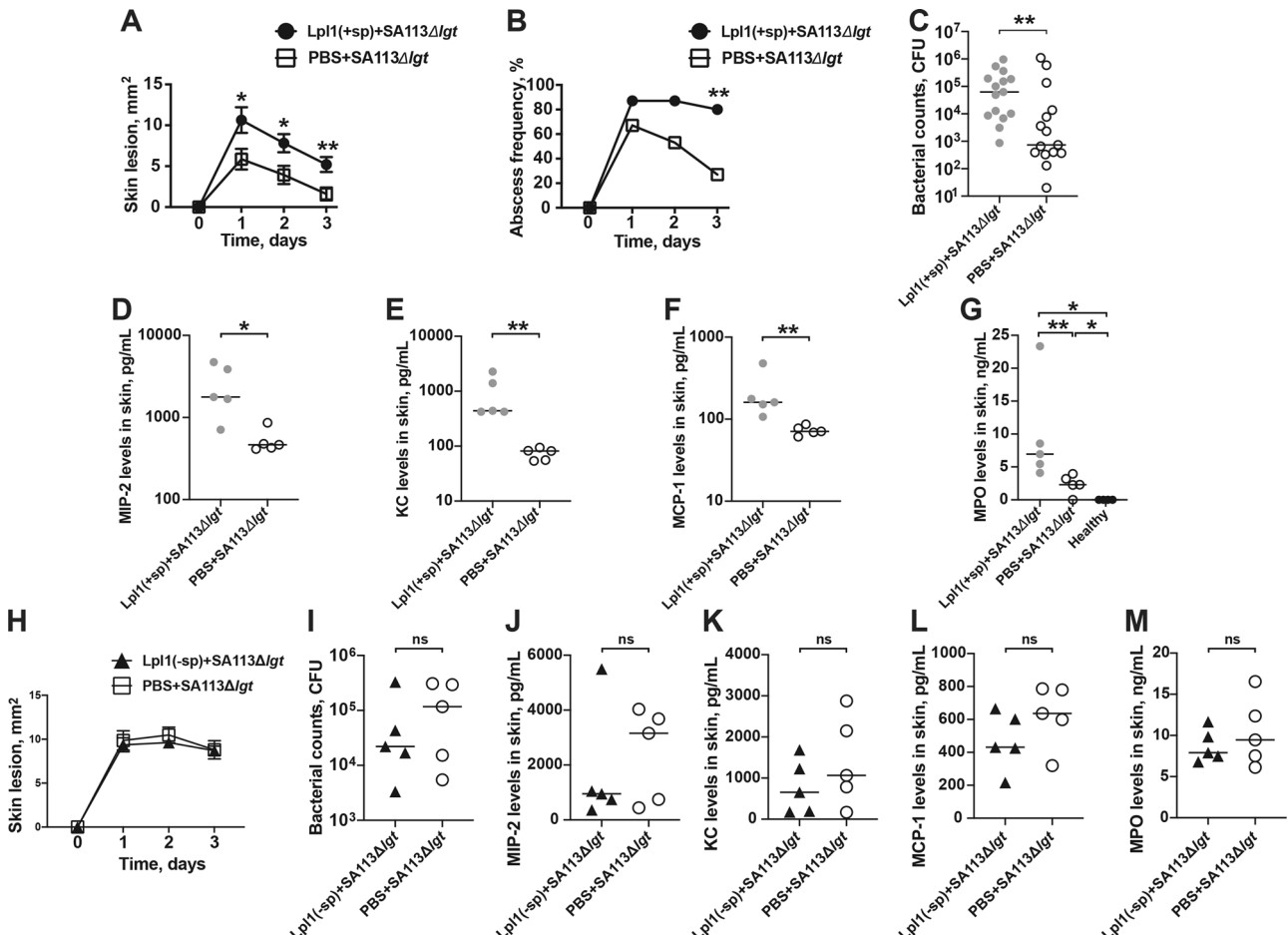

**Fig. 5 Coinjection of Lpp and live _S. aureus_ worsens the skin damage and increases the bacterial burden in local skin.** The skin lesion size (mm²) (**A**), and frequency of skin abscess formation (**B**) in NMRI mice ($n = 15$/group) up to 3 days after subcutaneous (s.c.) skin coinjection with 20 µl of _Staphylococcus aureus_ (_S. aureus_) SA113Δ_lgt_ mutant strain ($2.4 \times 10^6$ colony-forming units [CFU]/site) with either phosphate-buffered saline (PBS) or purified _S. aureus_ lipoprotein, denoted as Lpl1(+sp) (5 µg/site). Bacterial counts ($n = 15$/group) (**C**) and the levels of **D** macrophage inflammatory protein-2 (MIP-2), **E** keratinocyte chemoattractant (KC), **F** monocyte chemoattractant protein 1 (MCP-1), and **G** myeloperoxidase (MPO) ($n = 4$–5/group) in the supernatant of skin biopsy homogenates of the mice 3 days after s.c. skin infection of SA113Δ_lgt_ mutant strain ($2.4 \times 10^6$ CFU/site) with either PBS or Lpl1 (+sp) (5 µg/site). Data from three independent experiments with similar results were pooled (**A**–**C**). Samples from one representative experiment were used (**D**–**G**). The skin lesion size (mm²) development in NMRI mice ($n = 5$/group) up to 3 days (**H**) and bacterial counts (**I**), the levels of MIP-2 (**J**), KC (**K**), MCP-1 (**L**), and MPO (**M**) in the supernatant of skin biopsy homogenates of the mice 3 days after s.c. skin infection of SA113Δ_lgt_ mutant strain ($2.4 \times 10^6$ CFU/site) with either PBS or Lpl1(−sp) (5 µg/site). Statistical evaluations were performed using the Fisher's exact test (**B**), Mann–Whitney _U_ test, with data expressed as the mean ± standard error of the mean (**A**, **H**), or presented as scatterplot with line indicating median value (**C**–**G**, **I**–**M**). *$P < 0.05$; **$P < 0.01$; ns = not significant.

by a clear abscess capsule whereas the biopsies infected with NewmanΔ_lgt_ mutant strain showed distinctly less skin inflammation and capsule formation in the PBS-treated controls. More importantly, the histological sections showed no traces of any capsule formation in the skin biopsies from mice infected with the parental- or the Δ_lgt_ mutant strain in the fibrinogen-depleted group (Fig. 7J), indicating that the Lpp expressing bacteria are capable of forming a more isolated and protective environment by utilizing fibrinogen.

## Discussion

_S. aureus_ has a marked propensity to activate the coagulation system by producing coagulases[27]. Biological activities of coagulases substantially contribute to pathogenesis of _S. aureus_ infections including skin infection[28], septic arthritis[29], and sepsis[27]. It is known that staphylococci within fibrin meshwork are resistant to innate immune killing[27]. In the present study, we demonstrated that membrane anchored _S. aureus_ Lpps can also

manipulate local hemostasis toward procoagulant state by enhancing the expression of TF and PAI-1. Hypercoagulable state induced by Lpp promotes fibrin capsule formation, which protects bacteria from killing by immune cells. Our data show that Lpp-deficient _S. aureus_ displayed less virulence and was more susceptible to immune killing than its parental strain in the skin infection model. These results suggest an intriguing bacterial immune evasion mechanism mediated through Lpp expression by _S. aureus_.

In the skin tissue, there are several cell types contributing to skin immunity[30], including keratinocytes, Langerhans' cells, T cells, mast cells, dermal dendritic cells as well as macrophages. Subcutaneous injection of Lpp gave rise to a dose-dependent release of neutrophil and monocyte chemoattractants in local skin tissue, which consequently induced quick influx of monocytes and neutrophils but not NK or T cells. Occurrence of macrophages in skin tissue was also significantly increased. We speculated that tissue-resident macrophages were the main cell types producing chemokines upon Lpp

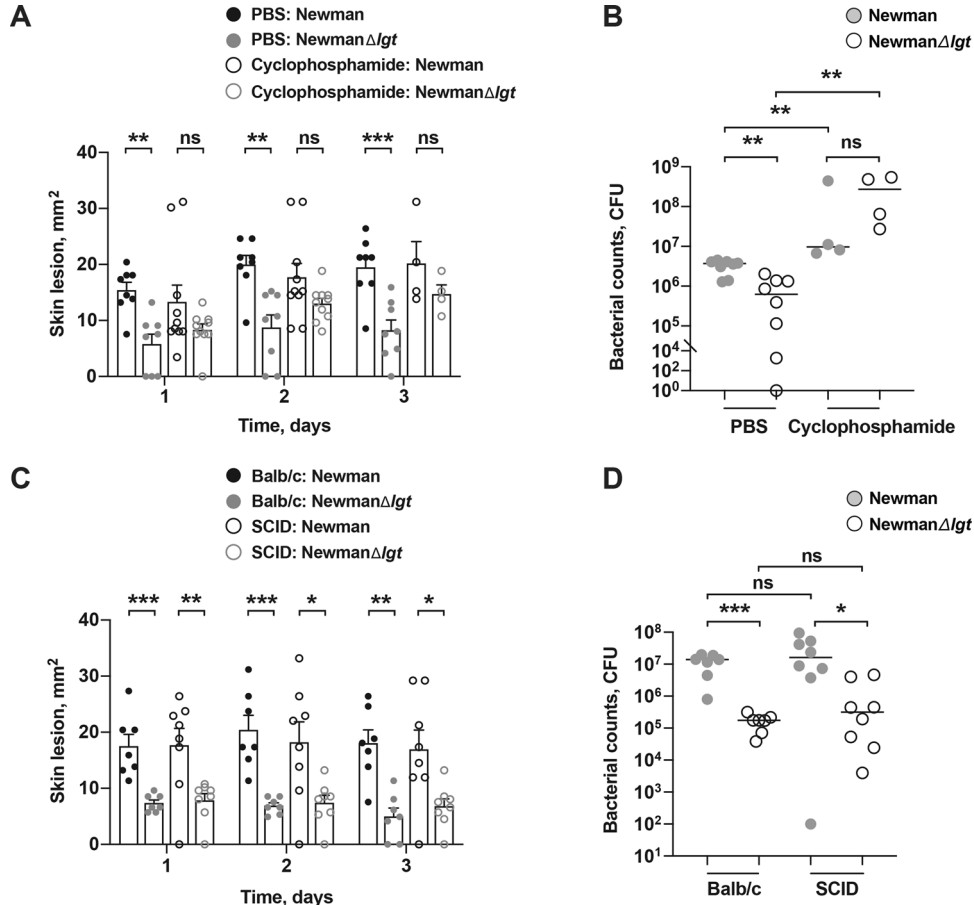

**Fig. 6 Difference in bacterial clearance between the parental- and Δ*lgt* mutant strain disappears by leukocyte depletion.** The skin lesion size (mm²) in mice ($n = 8$–$10$/group) up to 3 days (**A**) and bacterial counts in the supernatant of skin biopsy homogenates (**B**) from the surviving mice ($n = 4$–$8$/group) on day 3 after subcutaneous (s.c.) skin injection with 20 μl of *Staphylococcus aureus* Newman parental strain or NewmanΔ*lgt* mutant strain ($2 \times 10^6$ colony-forming units [CFU]/site) in NMRI mice depleted of leukocytes using cyclophosphamide or treated with phosphate-buffered saline (PBS) as control. The skin lesion size (mm²) up to 3 days (**C**) and bacterial counts in the supernatant of skin biopsy homogenates (**D**) on day 3 after s.c. skin infection with Newman parental strain or NewmanΔ*lgt* mutant strain ($2 \times 10^6$ CFU/site) in SCID mice or Balb/c controls ($n = 7$–$8$/group). The data were pooled from two independent experiments. Statistical evaluations were performed using the Mann–Whitney *U* test, with data expressed as the mean ± standard error of the mean (**A**, **C**), or presented as scatterplot with line indicating median value (**B**, **D**). *$P < 0.05$; **$P < 0.01$; ***$P < 0.001$; ns = not significant.

stimulation. Indeed, the importance of monocytes/macrophages in the inflammation induced by *S. aureus* bacterial components has been shown in the arthritis model[21,31–33]. Turnover of infiltrating monocytes to macrophages in the skin tissue resulted in the increased frequency of macrophages upon Lpp exposure, which further amplified the release of chemokines and consequent leukocyte infiltration. Not only macrophages, but also other cell types in skin tissue, including keratinocytes, fibroblasts, and endothelial cells are known to express these chemokines in vitro[34]. All these cells might also be another source for leukocyte chemoattractants responding to Lpp exposure in vivo.

Neutrophils, whose presence is increased in the skin at the site of Lpp injection, have proved to play an important protective role in *S. aureus* skin infections. Neutrophil-depleted mice inoculated with *S. aureus* developed chronic crusted ulcerations with a significantly higher burden of bacteria in the blood and skin[35]. *S. aureus*-induced IL-1β production by neutrophils is TLR2 dependent and known to be crucial for skin abscess formation[36]. As Lpp is the major ligand for TLR2 in *S. aureus* infection[22], Lpp deficiency results in fewer infiltrating neutrophils, less profound TLR2 activation, and consequently leads to reduced abscess formation. Indeed, abscess was hardly developed in leukopenic mice and Lpp effects on the skin lesion size and bacterial load were abrogated.

It has been shown that the vSaα specific *lpl* cluster of *S. aureus* facilitates bacterial invasion and initiation of skin infection[17]. In the current study, the importance of Lpp in established *S. aureus* skin infections became clear, as the skin lesion size was smaller in mice infected with *lgt*-deficient mutant strains compared with their parental strains. These effects were robust and observed in two mouse strains (both NMRI and C57BL/6 wild-type mice). In addition, both *agr* negative and *agr* positive *S. aureus* mutants (SA113Δ*lgt* and NewmanΔ*lgt*, respectively) displayed the similar phenotype, which strongly suggests that the pathogenicity of Lpp in *S. aureus* skin infections is not regulated by *agr* system. In contrast to the simplified model of purified Lpp injection where the TLR2 was obviously crucial for the host response to Lpp administration, the TLR2 dependency in the model of bacterial skin infections caused by live bacteria was less conclusive. In the TLR2 deficient mice, the lesion sizes between the Δ*lgt* mutant and parental strain were similar only at the early phase of the disease. The parental strain caused significantly bigger lesions later in the course of the disease, which was likely due to higher bacterial load. Two biological functions of Lpp are already known in *S. aureus* infections: (1) Lpp arouse a protective immune response through TLR2[18,20,21]; (2) Lpp are crucial for iron acquisition and better survival of bacteria, independent of host TLR2

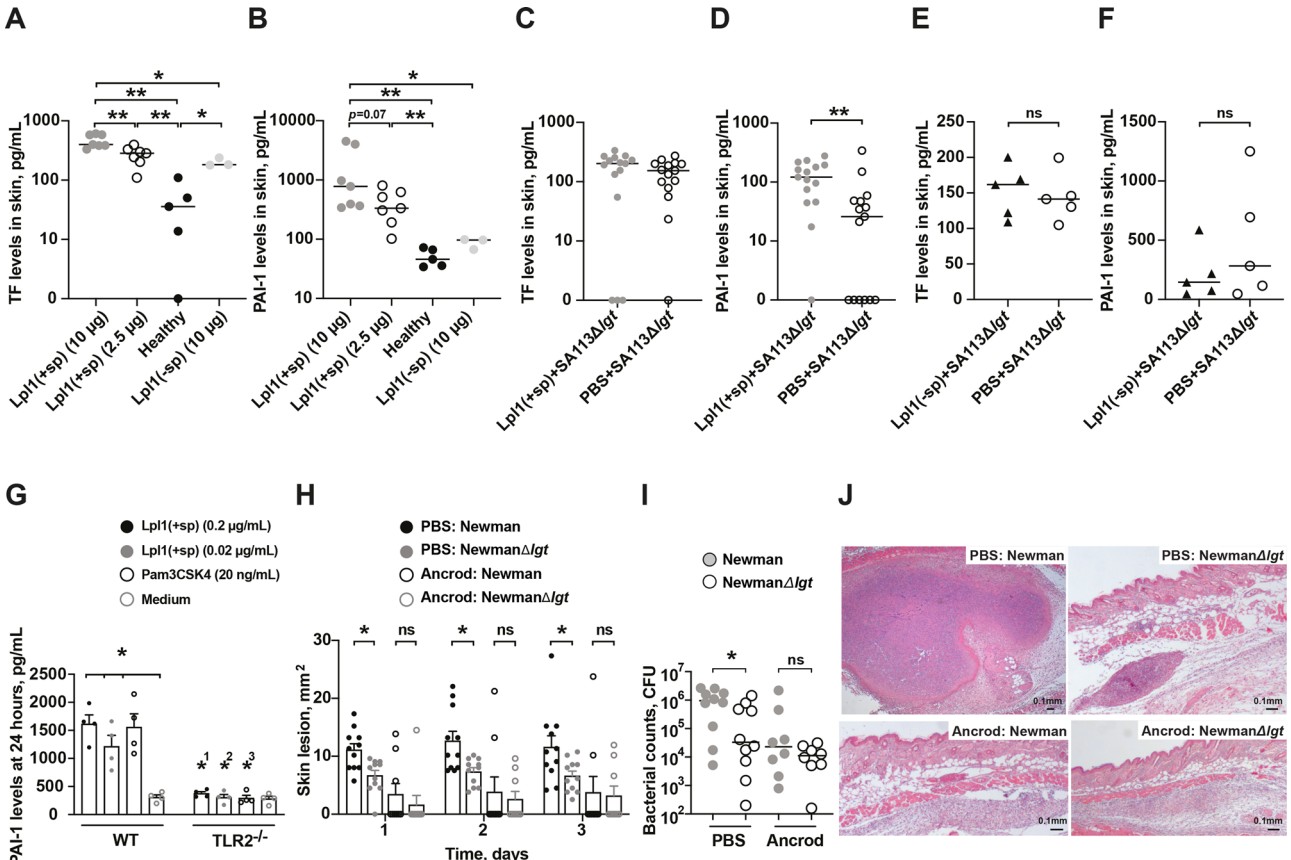

**Fig. 7 Lpp expression leads to an imbalanced coagulation/fibrinolysis hemostasis and fibrinogen depletion abrogates this Lpp-induced effect.** The levels of (**A**) tissue factor (TF) and (**B**) plasminogen activator inhibitor-1 (PAI-1) in the supernatant of healthy NMRI skin biopsy homogenates ($n = 5$), or in the skin biopsy homogenates on day 3 after subcutaneous (s.c.) skin injection with 20 μl of purified *Staphylococcus aureus* lipoprotein, denoted as Lpl1(+sp) (2.5 or 10 μg; $n = 7$/group) or unlipidated Lpl1 protein, denoted as Lpl1(−sp) (10 μg; $n = 3$) in NMRI mice. The levels of **C** TF and **D** PAI-1 in the supernatant of skin biopsy homogenates from NMRI mice 3 days after s.c. skin coinjection with 20 μl of SA113Δ*lgt* mutant strain ($2.4 \times 10^6$ colony-forming units [CFU]/site) with either phosphate-buffered saline (PBS) or Lpl1(+sp) (5 μg/site; $n = 15$/group). The levels of **E** TF and **F** PAI-1 in the supernatant of skin biopsy homogenates from NMRI mice 3 days after s.c. skin coinjection with 20 μl of SA113Δ*lgt* mutant strain ($2.4 \times 10^6$ CFU/site) with either PBS or Lpl1(−sp) (5 μg/site; $n = 5$/group). **G** PAI-1 levels in the supernatants collected from C57BL/6 wildtype (WT) and TLR2 deficient (TLR2$^{-/-}$) ($n = 4$/group) mouse peritoneal macrophage cell cultures ($5 \times 10^5$ cells/mL) after stimulation for 24 h with Lpl1(+sp) (0.02 or 0.2 μg/mL), Pam3CSK4 (20 ng/mL), or culture medium. NMRI mice depleted of fibrinogen using Ancrod or treated with PBS as control and s.c. infected with 20 μl of *Staphylococcus aureus* Newman parental strain or NewmanΔ*lgt* mutant strain ($2 \times 10^6$ CFU/site) to assess **H** the skin lesion size (mm²) up to 3 days postinfection ($n = 9$–11/group), and **I** the bacterial counts in the supernatant of skin biopsy homogenates on day 3 postinfection ($n = 8$–10/group). Representative photomicrographs (**J**) of mouse skin lesions in NMRI mice s.c. infected with Newman parental strain or NewmanΔ*lgt* mutant strain treated with either PBS (left upper panel and right upper panel, respectively) or Ancrod (left lower panel and right lower panel, respectively) on day 3 postinfection, stained with hematoxylin and eosin. Original magnification ×2.5. The data were pooled from 2 to 3 independent experiments. Statistical evaluations were performed using the Mann–Whitney *U* test, with data presented as scatterplot with line indicating median value (**A**-**F**, **I**), or the mean ± standard error of the mean (**G**-**H**). $^{1,2,3}$ = Comparisons of corresponding stimuli between WT and TLR2$^{-/-}$ mice. *$P < 0.05$; **$P < 0.01$; ns = not significant.

expression[13],[18]. The clinical outcome of an infection is, therefore, an overall result of these double effects induced by Lpp. Since differences in bacterial loads were TLR2 independent, the metabolic fitness or other TLR2 independent mechanisms of Lpp are most likely prodominent in model of *S. aureus* skin infections. Also, IL-1R/MyD88 rather than TLR2/MyD88 signaling in resident skin cells was shown to be crucial for promoting neutrophil recruitment in *S. aureus* skin infections[37], which gives another explanation to the TLR2 independency of Lpp in skin infections. In general, TLR2 possesses a protective function during *S. aureus* infection[20],[38],[39]. Notably, we found no increase in bacterial burden in TLR2 deficient mice compared to wild-type mice. This finding corroborates that of Miller et al.[37] but is in contrast to that of Hoebe et al.[40]. We speculate that the discrepant roles of TLR2 in different studies might be due to different bacterial strains used and different infectious doses administered in the skin.

It is clear that in *S. aureus* skin infection Lpp expression protects the bacteria from immune killing. As mentioned above, Lpp mediated metabolic fitness might be one of the explanations for better survival of *S. aureus* in the skin tissue. However, addition of exogenous purified Lpl1 to *lgt* mutant strain, which has no impact on the bacteria metabolic fitness, also increased the skin lesion size and bacterial burden in the skin. In addition, leukocyte depletion led to total abrogation of effect induced by Lpp in terms of bacterial load. All these evidence suggest that Lpp has some other unknown biological functions that contribute to the bacterial survival and persistence of skin infection.

The importance of a balanced coagulation/fibrinolysis hemostasis in *S. aureus* infections has been shown before[41]. Our current data demonstrate that Lpp expression by *S. aureus* manipulates hemostasis toward hypercoagulable state and local fibrin deposition, which may contribute to abscess formation. Clearly,

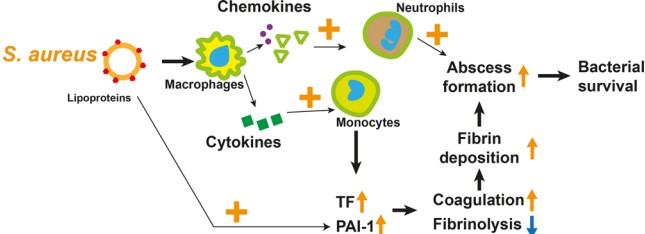

**Fig. 8 Schematic diagram of proposed new function of *S. aureus* lipoproteins in skin infection.** *Staphylococcus aureus* (*S. aureus*) lipoproteins stimulate the release of chemokines and cytokines from macrophages, consequently leading to substantial influx of monocytes and neutrophils to the site of infection. In turn, this enhances the levels of tissue factor (TF) and plasminogen activator inhibitor-1 (PAI-1), initiating the activation of the coagulation and inhibition of fibrinolysis, which results in deposition of fibrin and increased skin abscess formation with improved bacterial survival. In addition, neutrophils play a crucial role in directly inducing abscess formation.

*lgt*-deficient mutant strain is less competent to induce skin abscess formation compared to its parental strain. Skin abscess formation is a very efficient method of the host to limit an infection in a capsule from an uncontrolled spreading. However, since abscess wall cuts off the possibility for innate immune cells to migrate to the infection site, bacteria inside of an abscess are not accessible to innate immune killing and, therefore, survive better. We have shown before that opening and draining of abscesses due to plasminogen activation by staphylokinase expressed in *S. aureus* decreased severity of skin infections[42]. Fibrin is one of the major components of capsules and pseudo-capsules in abscesses[43]. Moreover, fibrin scaffold was shown to shield *S. aureus* from opsonophagocytosis and innate immune attack[44]. TF initiates activation of coagulation cascade during infections resulting in deposition of fibrin[26]. Monocytes/macro-phages are efficient TF producers[26]. As monocytes quickly infil-trate to the Lpp injected skin tissue where the environment is rich by these proinflammatory cytokines, it is no surprise that TF expression was upregulated in the local tissues. Simultaneously, PAI-1, the potent inhibitor of fibrinolytic system, was greatly upregulated in the skin tissue by subcutaneous injection of Lpp, which further enhanced the local fibrin deposition and abscess capsule formation. Indeed, mice depleted of fibrinogen failed to form the typical skin abscess. Importantly, the differences in lesion size and bacterial burden between the *lgt*-deficient strain and parental strain disappeared in the fibrinogen-depleted mice, suggesting that effects induced by Lpp expression in *S. aureus* skin infection are related to expression of host fibrin/fibrinogen. The new function of *S. aureus* Lpp in *S. aureus* skin infection is summarized in a schematic diagram (Fig. 8).

In summary, our data strongly suggests that Lpps play a potent role in *S. aureus* skin infections. Lpp expression in *S. aureus* leads to higher bacterial load and more severe skin lesions. Such effect might be due to enhanced skin abscess formation caused by imbalanced coagulation/fibrinolysis hemostasis by Lpp in the local skin tissue, which is a new bacterial evasion mechanism.

## Methods

**Ethics statement**. Mouse studies were reviewed and approved by the Ethics Committee of Animal Research of Gothenburg. Mouse experiments were con-ducted in accordance with recommendations listed in the Swedish Board of Agriculture's regulations and recommendations on animal experiments.

**Mice**. Female NMRI mice, aged 6–10 weeks, were purchased from Envigo (Venray, Netherlands), gender- and age-matched 6- to 10-week-old C57BL/6 wild-type mice and Toll-like receptor 2-deficient B6.129-Tlr2^tm1Kir/J (TLR2⁻/⁻) mice were

purchased from Charles River Laboratories (Sulzfeld, Germany) and The Jackson Laboratory (Bar Harbor, Maine, USA), respectively. Female CB17-SCID mice and Balb/c mice, aged 8 weeks, were purchased from Charles River Laboratories (Sulzfeld, Germany). All mice were housed in the animal facility of the Department of Rheumatology and Inflammation Research, University of Gothenburg. Mice were kept under standard temperature and light conditions and were fed laboratory chow and water ad libitum. The Ethics Committee of Animal Research of Gothenburg approved the study, and the animal experimentation guidelines were strictly followed.

**Expression and purification of Lpl1(+sp) and Lpl1(−sp)**. The preparation and purification of the *S. aureus* lipoproteins Lpl1(+sp) and Lpl1(−sp) were performed by Dr. Nguyen (Microbial Genetics, University of Tübingen, Germany), as pre-viously described[17]. Lpl1(+sp) was isolated from the membrane fraction of *S. aureus* SA113 (pTX30::*lpl1*-his), whereas Lpl(−sp) was isolated from the cyto-plasmic fraction of *S. aureus* SA113Δ*lgt* (pTX30::*lpl1*(−sp)-his), as described elsewhere[21]. Finally, the Lpl1-his purification was checked by SDS-PAGE, as previously described[17]. The purified compounds of Lpl1 were stored at −70 °C until use and were adjusted in PBS to the required concentration before each experiment.

**Bacterial strains and preparation of bacterial solutions**. The SA113 parental strain, and its lipoprotein-deficient Δ*lgt* mutant strain, SA113Δ*lgt* mutant[16] *S. aureus* strain were prepared as described. The *S. aureus* complemented mutant strain, SA113Δ*lgt* (pRB474::*lgt*) was prepared as described[16]. The *S. aureus* New-man parental strain and its NewmanΔ*lgt* mutant strain were generated exactly as previously described[16]. The *S. aureus* Newman parental strain and NewmanΔ*lgt* mutant strain were cultured separately on horse blood agar plates or trypticase soy agar (TSA) plates with erythromycin (2.5 µg/ml), respectively for 24 h and stored as previously described[20]. The bacterial solutions were thawed, washed with sterile PBS, and adjusted to the required concentration before each experiment.

**Experimental protocols for murine *S. aureus* skin infection and *S. aureus* lipoprotein induced skin inflammation**. To study the effect of *S. aureus* Lpp in murine skin inflammation and skin infection, seven sets of experiments were performed using NMRI, C57BL/6 wild-type, TLR2⁻/⁻, SCID or Balb/c mice. The animals were anaesthetized with ketamine hydrochloride (Pfizer AB, Sweden) and medetomidine (Orion Pharma, Finland), their backs were shaved, and s.c. injected with one of the following compounds, or bacterial solutions, adjusted to the desired concentration in PBS: (1) 20 µl of purified Lpl1(+sp) or Lpl1(−sp) *S. aureus* Lpp; (2) 50 µl of live SA113, SA113Δ*lgt* mutant or complemented mutant SA113Δ*lgt* (pRB474::*lgt*) *S. aureus* strains; (3) 20 µl of live Newman or NewmanΔ*lgt* mutant *S. aureus* strains; (4) 20 µl of solutions containing mixtures of live SA113Δ*lgt* mutant *S. aureus* strain together with either Lpl1(+sp), Lpl1(−sp) or PBS; (5) 20 µl of live Newman or NewmanΔ*lgt* mutant *S. aureus* strains in mice depleted of leukocytes; (6) 20 µl of live Newman or NewmanΔ*lgt* mutant *S. aureus* strains in SCID mice and Balb/c controls; (7) 20 µl of live Newman or NewmanΔ*lgt* mutant *S. aureus* strains in mice depleted of fibrinogen. In addition, 10 µl of purified Lpl1(+sp) *S. aureus* Lpp or PBS were injected s.c. into C57BL/6 wild-type or TLR2⁻/⁻ mouse auricle and ear tissues were then collected for immune cell phenotyping.

The resulting skin lesions were measured with calipers until the mice were sacrificed. The sizes of the skin lesions were calculated using the mathematical formula for the area of an ellipse. Two observers (M.M. and T.J.) inspected the lesion size of each mouse in a blinded manner. Presence of cutaneous abscess was defined as the localized collection of pus or translucent fluid within the skin and/or below the skin surface with clinical firm or fluctuant swelling. A healed skin lesion was defined as a skin damage fully recovered, lacking any clinical signs of inflammatory exudates, or skin abscess. At the end of experiments, the mice were anaesthetized with ketamine/medetomidine, the mice were sacrificed by a cervical dislocation, and skin biopsies was collected afterward.

**Immune cell phenotyping of skin tissue using flow cytometry**. A dose of 1 µg of Lpl1(+sp) in 10 µl of PBS or PBS alone as internal control, were s.c. injected into the auricle of anesthetized C57BL/6 wild-type (*n* = 6), and TLR2⁻/⁻ (*n* = 5) mice. On day 1 after injection, the ear tissues were collected, placed in RPMI medium (Fisher Scientific), and subjected to enzymatic digestion with 4 mg/ml Collagenase IV (Fisher Scientific) and 0.4 mg/ml DNase I (Sigma-Aldrich) in RPMI medium, followed by incubation for 1 h at 37 °C with shaking. A single-cell suspension was obtained after the tissue was homogenized and passed through a 40 µm cell strainer (Becton Dickinson). Skin tissue cells were then analyzed using the following antibodies: APC-R700-conjugated anti-mouse CD45 (BD Biosciences), V450-conjugated anti-mouse CD11b (BD Biosciences), APC-conjugated anti-mouse F4/80 (BioLegend), PE-Cy7-conjugated anti-mouse Ly-6G (BD Biosciences), BV605-conjugated anti-mouse Ly-6C (BD Biosciences), PerCP-Cy5.5-conjugated anti-mouse CD335 (BD Biosciences), and FITC-conjugated rat anti-mouse CD3 (BD Biosciences). Cells were acquired on a BD FACSLyric flow cytometer (BD Bios-ciences) and data were analyzed using FlowJo version 10.1 software (Tree Star, Ashland, USA).

**In vivo leukocyte depletion procedure**. NMRI mice were intraperitoneally injected with either 200 µl of PBS as a control substance, or with 450 mg/kg body weight of cyclophosphamide (Sigma-Aldrich, Saint Louis, MO, USA) in 200 µl of PBS/mouse, four days and one day prior to the skin infection as previously described[42]. The experiment was terminated on day 3 postinfection and the level of peripheral mouse blood leukocytes were thereafter assessed in a cell counter (Sysmex, Kobe, Japan), which remained <2.5% of the normal count (Supplementary Fig. 7).

**In vivo fibrinogen depletion procedure**. Ancrod (Product No. 15/106; National Institute for Biological Standards and Control, South Mimms, UK), a thrombin-like enzyme derived from the snake venom of the Malayan pit viper *Calloselasma rhodostoma*, is known to efficiently deplete fibrinogen levels in vivo[41,45,46]. NMRI mice were intraperitoneally injected with either 200 µl of PBS as a control substance, or with 2 units of Ancrod in 200 µl of PBS/mouse every 12 h. The treatment started 12 h prior to the skin infection and continued until the experiment was terminated on day 3 postinfection.

**Skin homogenate preparation and bacteriologic examination of skin biopsies**. On days 3 and 10 postinfection, the mice were euthanized, the skin was disinfected with 70% v/v ethanol, and the skin biopsies encompassing the entire inflamed or infected area were taken with a sterile 8-mm biopsy punch (Kai Medical, Seki, Japan), as previously described[47], and afterward homogenized with TissueLyser II (Qiagen, Hilden, Germany) in 0.5 ml PBS. Thereafter, 100 µl from the homogenates were further diluted in PBS, spread on horse blood agar plates, and incubated for 24 h at 37 °C. Viable counts of bacteria were performed and quantified as CFUs. This method recovers approximately 85% of the bacteria present in a skin sample[48]. The remaining volume of skin homogenates were spun at 13,000 rpm for 10 min. The supernatant was collected and subsequently used for measurement of chemokines and myeloperoxidase.

**Measurement of chemokine and myeloperoxidase levels in skin homogenates**. The level of myeloperoxidase was measured using a mouse MPO ELISA kit (Thermo Fisher Scientific, Waltham, MA, USA) and the levels of MIP-2, KC, MCP-1, TF, and PAI-1 in the supernatants from skin homogenates were quantified using DuoSet ELISA Kits (R&D Systems, Abingdon, UK) according to manufacturer´s instructions.

**In vitro stimulation of peritoneal macrophages**. Peritoneal macrophages from healthy C57BL/6 wild-type and TLR2$^{-/-}$ mice ($n = 4$/group) were prepared and performed as previously described[21]. Briefly, a density of $5 \times 10^5$ cells/mL were used and the cells were stimulated with purified Lpl1(+sp) (0.02 or 0.2 µg/mL), Pam3CSK4 (20 ng/mL) (EMC, Tübingen, Germany), or culture medium. The supernatants were collected after 24 h, and the levels of TF and PAI-1 were later analyzed.

**Histopathological examination**. Skin biopsy samples collected on day 3, were fixed with 4% phosphate-buffered formaldehyde, embedded in paraffin and sectioned with a microtome. Tissue sections were thereafter stained with hematoxylin and eosin. All slides were coded and assessed under a microscope in a blinded manner by two observers (T.J. and M.M.).

**Statistics and reproducibility**. All statistical analyses were performed using GraphPad Prism version 8.0.2 software for Macintosh (GraphPad Software, La Jolla, CA, USA). Statistical significance was assessed using the Mann–Whitney U test, Fisher's exact test, and Mantel–Cox log-rank test as appropriate. The results are reported as the mean ± standard error of the mean (SEM), or the median unless indicated otherwise. A $p$ value < 0.05 was considered statistically significant. Numbers of repeats for each experiment were described in the associated figure legends.

**Reporting summary**. Further information on research design is available in the Nature Research Reporting Summary linked to this article.

## Data availability
The authors declare that the main data supporting the findings of this study are available within the article and its Supplementary files. Source data underlying plots shown in figures are provided in Supplementary Data 1–7. Extra data are available from the corresponding author upon request.

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

## Acknowledgements

This work was supported by the Swedish Medical Research Council [grant number 523-2013-2750 to T.J.]; grants from the Swedish state under the agreement between the Swedish Government and the county councils, the ALF-agreement [grant number ALFGBG-823941 to T.J., ALFGBG-770411 to A.J., ALFGBG-926621 to R.P.]; Professor Nanna Svartz Fond [grant number 2016-00117 to T.J., 2014-00058 and 2016-00154 to R.P.]; the Swedish Rheumatism Association [grant numbers R-385441, R-478421 to R.P.]; the Swedish Medical Society [grant number SLS-505901 to R.P.]; the Gothenburg Society of Medicine [grant number GLS-784641 to A.J.]; the Wilhelm and Martina Lundgren Foundation to [T.J., M.N., A.A., A.J., and R.P.]; Rune och Ulla Amlövs Stiftelse för Neurologisk och Reumatologisk Forskning [grant number 2016-075 to T.J., 2015-00053, 2017-129 to A.J.]; Adlerbertska Forskningsstiftelsen to [M.M., A.A., M.N., A.J., and T.J.]; Inger Bendix stiftelse to [A.J.]; Sahlgrenska University Foundations to [A.J.]; Kungl. Vetenskapsakademiens stiftelser [grant number ME2015-0119 to A.A.]; E och K.G. Lennanders stipendiestiftelse to [M.M. and A.A.]; Magnus Bergvalls Stiftelse [grant numbers 2017-01958, 2018-02797 to A.A.] and Institute of Medicine, Gothenburg University. M.T.N. and F.G. were funded by the Deutsche Forschungsgemeinschaft (DFG) Germany´s Excellence Strategy—EXC 2124—390838134 "Controlling Microbes to Fight Infections". The funders had no role in study design, data collection and analysis, decision to publish, or preparation of the manuscript.

## Author contributions

M.M. and T.J. conceived the study. M.M. and T.J. designed the experiments. M.M. performed most of the experiments, and M.M., M.N., Z.H., P.K., A.J., and A.A. carried out the remaining data collection, and statistical analysis of the data. The manuscript was drafted by M.M. and T.J., and critically revised by M.M., M.N., Z.H., M.T.N., P.K., A.J., A.A., A.K., R.P., F.G., and T.J. All authors have read and approved the final version of the manuscript.

## Funding

## Competing interests

The authors declare no competing interests.
