## [Peer Review File · Communications Biology]

Reviewers' comments:

Reviewer #1 (Remarks to the Author):

In “Staphylococcus aureus lipoproteins induce skin inflammation and promote abscess formation, shielding bacteria from immune killing”, Majd Mohammad and colleagues examine immune effects of lipoproteins of the human pathogen Staphylococcus aureus. They show that lipoproteins as proinflammatory, and bacteria deficient in them are less inflammatory. These observations are not entirely novel, but of interest and potential importance. However, key experiments are lacking adequate controls and do not fully support the conclusions. Specific notes follow:

Major comments

Altogether, significance appears to be usually reached based on animal number, and not necessary biological mechanism. Power functions need to be used to determine appropriate sample numbers which should be applied more consistently. In some experiments large numbers of mice are used to make fairly minor points (eg 32 mice in Fig 5C), while others are very underpowered (Figure 2 makes conclusions on the importance of TLR2 based on groups of 3 mice). Many appropriate comparisons appear that they would reach significance with adequate mouse numbers (for example, monocyte and macrophage number induced changes induced by Lpl1 in TLR2 mice, which does not agree with the proposed molecular mechanisms). In many paired datasets, the number of animals changes, suggesting samples are being omitted or other factors are confounding analysis. Its also not clear how many experiments are performed, when samples are combined, or are representative, or these differences decided.

Lpp deficient bacteria, expectedly, have decreased viability and growth, for reasons including deficiencies in iron acquisition. These mutants are more susceptible to killing by host immune effectors and are rapidly cleared – any mutant that so fundamentally impacts bacterial survival will decrease inflammation, but this is very non-specific and not a mechanism. This has not been sufficiently discussed or deconvolved as a contributor to the in vivo observations. The attenuation of Lgt mutants also appears highly variable between strains and experiments. Prior literature is all consistent with these mutants being highly attenuated, which makes sense, but why this is seen in some experiments and not others is of concern. Potentially there are issues with mutants, or these are not fundamental mechanisms, but a phenomenology of certain strains that would at least require additional explanation.

Line 147, it is not clear that the wound healing process was “faster”, as stated. The data are consistent with healing starting sooner, because the infection is resolved soon, since the bacteria are attenuated. Line 176; similar comment. The apparent role of lpp early in infection is likely due to attenuation and reduced bacterial burden; evidence is not consistent with roles of TLR2 signaling in this. The slopes of lesion size (figure 4, for example) suggest wound healing is not different, just starting from a different size.

Line 216. Also consistent with this model is that cyclophosphamide depletion of important immune cells reduces the killing of staph, even leading to a partial attenuation reversal of the lpp mutant strain.

Fibrin is a barrier to bacterial dissemination; therefore, ancrud depletion of fibrinogen is expected to

promote staph dissemination. Figure 7 shows smaller lesions with ancred treatment, with correspondingly fewer bacteria, but it is not clear that there just isn't fewer bacteria due to spread to other sites.

Minor comments

Minor edits for grammar needed throughout

Greater discussion on the number of Lpp proteins and their functions is needed

No details on purity of Lpl1

Line 167, discussion on the significance of agr+ would be useful

Are cytokines normalized by lesion size? Methods on sample collection could use more detail. Cytokines, for example, on Y-axis of graphs appear to be normalized to tissue mass. However, methods indicate skin was consistently collected by 8mm biopsy punch. This would be irrespective of lesion size, so small lesions will inherently have the cytokine numbers diluted out by normal tissue.

Histology lacks scale bars

Reviewer #2 (Remarks to the Author):

This study shows that the lipid moiety of *S. aureus* lipoprotein (Lpp) contributes to the pathogenesis of skin infection by stimulating inflammation and abscess formation, using purified Lpp (Lpl1) and *S. aureus* lacking lipidated Lpp (Δ lgt mutant) in mice. The authors also suggest that Lpp-induced abscess formation contributes to staphylococcal immune evasion given that fibrinogen depletion promoted WT *S. aureus* clearance but not the Δ lgt mutant. While interesting, these are not novel concepts (Schmaler et al. *J Immunol* June 1, 2009, 182 (11) 7110-7118, Cheng et al., *FASEB J.* 2009 Oct; 23(10): 3393–3404). I have the following comments:

1. The interpretation of data in the manuscript is often confusing or at times not quite accurate (see point 4 below please). The discussion section should be more concise.
2. The inconsistency in the role of TLR2 stimulation by Lpp upon the use of purified Lpp (Lpl1) versus during bacterial infection was not clearly explained. While Lpl1-induced skin lesion was dependent on TLR2, the Δ lgt mutant was attenuated in both the WT and TLR2^{-/-} mice. The latter observation could be due to the fact that the Δ lgt mutant has a growth defect under nutrient limitation compared to the parent strain as previously shown by Stoll et al. (*Infect Immun* 2005, 73(4):2411-2423) and the authors (Mohammad et al., *Scientific Reports* 2020, 10: 7936).
3. The authors show that during skin infection TLR2 does not control *S. aureus* infection (Fig. 4C, no increase in bacterial burden in TLR2^{-/-}-infected mice) but fail to mention and discuss this observation which corroborates that of Miller et al. (*Immunity* 2006, 24:79-91) but is in contrast to

that of Hoebe et al. (Nature 2005, 433:523-527). The differing roles of TLR2 in skin infection have been suggested to be due to different bacterial strains used and different infectious doses administered in the skin. What doses of bacteria were given to the mice? The methods section only mention bacterial volumes.

4. (Page 10 lines 199-217) Paragraph title "The effect induced by Lpp expression is abrogated by leukocyte depletion". This paragraph focuses on the attenuation of the Δ lgt mutant compared to the parent strain depending on the presence of leukocytes when in fact what this shows is that leukocytes are important to contain *S. aureus* skin infection.

5. Line 206 - "cyclophosphamide-treated mice developed more severe disease", can you please show the statistical significance in CFUs from cyclophosphamide-treated versus untreated mice (Fig. 6B)?

6. Did you verify leukocyte depletion upon cyclophosphamide-treatment? (e.g. Flow data?)

7. Lines 207-208 - "The leukopenic mice also lost significantly more weight compared to ...infection". No figure callout, where are the data?

Minor comments:

1. Line 49 "unique structure" of *S. aureus*, please rephrase

2. Line 65 *S. aureus* 'contributes', line 350 Lpp 'plays'

3. Line 120 Lpp 'induces'

4. Why was the immune cell analysis done in infected mouse auricles and not from the skin biopsies (injected mouse backs as per methods)?

5. Line 161-162 - "SA113 Δ lgt mutant strain tended to induce lower levels of MPO in the skin tissue than SA113 parental strain (Fig. 3D). But there was no significant difference according to that figure?"

6. Lines 337-338 - "Bacterial MAMPs like LPS", what is the relevance to *S. aureus*?

7. Fig. 5G - MPO levels much lower compared to Fig. 3D, do these fluctuate that much?

Reviewer #3 (Remarks to the Author):

Summary

In this study, authors showed Lipoprotein(Lpp) is necessary to recruit leukocytes and induce inflammation through TLR2 signaling. They also showed *S. aureus* with Lpp enhances skin abscess formation caused by imbalanced coagulation/fibrinolysis hemostasis in the local skin tissue by using Lpp deficient strain(Δ lgt). It is interesting idea that abscess formation is the bacterial evasion mechanism. But there is important question how does host immune function fight against Lpp deficient bacteria without innate immune response? Since Newman Δ lgt has slow growth rate in biological conditions (Mohammad, M. et al. The role of Staphylococcus aureus lipoproteins in hematogenous septic arthritis. Scientific Reports 10, 7936, doi:10.1038/s41598-020-537 64879-4 (2020)), it would be difficult to simply compare the number of bacteria after 3 or 10 days. Also, authors use two different strains, SA113(Agr +) and Newman(Agr -), in different experiment. What is the impact from Agr system in this phenotype? It would be better to clarify in result or discussion.

Overall impression

The idea that bacteria take advantage of abscess formation to survive in host homeostasis, and their strategy is straight forward. But they need to address those questions.

Comments/Suggestion for authors

Major Queries

1, In figure 3 authors showed the SA113Δlgt strain cause smaller infectious lesion and less bacterial counts (Figure 3C). It looks small difference and is nuclear the original number of injected SA113. Figure 3D shows SA113Δlgt has similar MPO level to WT strain. But in Figure 4B and D, NewmanΔlgt strain still cause smaller infectious lesion and less bacteria even in TLR2^{-/-} mice. Furthermore, in figure 6, Cyclophosphamide treated leukocyte depletion mouse injected Newman parental strain didn't show higher larger skin lesion compare to PBS treated mice. Taken together it suggest Lpp is necessary to cause inflammation but bacterial survival or overload is due to the different growth speed of each bacteria. In other words, the low number of bacteria is caused by slower growth of Δlgt strain.

Could you address how does host fight against Lpp deficient bacteria without innate immune response? Is there any difference in adaptive immunity? Or is it possible to knock in Lpp in Newman or SA113 Δlgt strain fo repeat experiment?

2, In figure 4, Authors use Newman(Agr -) strain instead of SA113(Agr +) for TLR2 Ko mice experiment. But there is no discussion about the impact of Agr system. What is the impact of Agr system?

Minor Queries

1, The authors showed CFU in Figure 3D, 4C, 5C, 6B and 7G, but those are hard to see the difference even though there is statistically significant difference.

Reviewer #4 (Remarks to the Author):

Staphylococcus aureus lipoproteins (Lpp) are membrane anchored surface proteins that play important roles in host-bacterial interactions. How Lpp involved in S. aureus pathogenicity remain poorly studied. Mohammad and colleagues showed in their previous study that Lpp induces chronic destructive macroscopic arthritis (PMID: 31226163). In this study, Mohammad and colleagues further investigate the role of Lpp, in particular "lipoprotein-like" Lpl1, in murine skin infection. They find that subcutaneous injection of Lpl1 promote leukocytes infiltration and skin lesion, which are TLR2-dependent. Also, in comparison to wild type strain, lgt-mutant strain causes smaller skin lesion size and lower bacterial loads, which the latter is TLR2-independent. Finally, they find that the skin lesions and bacterial burden induced by Lpp can be abrogated by chemically-depletion of leukocyte or fibrinogen.

This study is potentially interesting, particularly lpp-induced fibrin capsule and abscess formation as a strategy to evade host immune attack, and will be of interest to a broad audience of infectious disease researchers and microbiologists. However, the mechanism underlying the abscess formation to protect S. aureus from immune cells is less convincing, as the conclusion was made largely based on experiments using purified Lpl1 alone or co-injection with live bacteria. More rigorous examinations are required in this regard.

Below are comments/points that may help to improve the manuscript and hope the authors will find them useful.

Major points:

1. The authors showed increased chemoattractant level after purified Lpl1 s.c. injection (Fig 1), thus promoting the infiltration of leukocytes to the injection site (Fig 2). However, the experiments heavily rely on purified Lpl1 treatment, an approach that may undermine other Staph virulence factors that also cause skin inflammation and leukocytes infiltration. It would be great if the authors could use lgt-mutant and complemented mutant lgt strains to validate the significant impact of Staph Lpp in chemoattractant release and leukocytes infiltration at the skin infection site.

2. In general, TLR2 possess a protective function during *S. aureus* infection, which involves in neutrophils recruitment in response to *S. aureus*. The bacterial burden in the murine organs (Takeuchi et al. 2000, *J Immunol*, PMID: 11067888; Yimin et al. 2013, *Plos One*, PMID: 24058538) and skin (Miller et al. 2006 *Immunity*, PMID: 16413925) were higher in TLR2-deficient mice than wild type mice. In contrast, this study show a smaller skin lesion size and normal bacterial clearance that is similar to wild type mice (Fig 4). The authors should discuss this discrepancy.

3. What are the levels of MIP-2, KC and MCP-1 of the skin biopsy homogenate in TLR2^{-/-} mice after Lpl1 injection, as well as in WT vs TLR2^{-/-} mice after WT and lgt-mutant strains infection? It would be interesting to see whether these chemoattractant releases are dependent on TLR2.

4. Despite the possibility of increased neutrophils and monocytes infiltration, higher bacterial burden was observed on day 3 after s.c. skin infection of both wild type SA113 and Newman strains compared to lgt-mutant strains (In Fig 3C and 4C). It is not convincing that lgt-mutant burden is significantly lower than wild type strain at day 3 as the difference is marginal (less than 2-fold). The authors also suggest that Staph Lpp provoked less bacterial clearance in wild type SA113 compared to lgt-mutant strain (Line 156 – 158). However, Stoll et al. 2005, *Infect Immun* (PMCID: PMC1087423) and Bubeck Wardenburg et al. 2006, *PNAS* (PMCID: PMC1564215) demonstrated that the growth of lgt-mutant strain was retarded under nutrient limitation and stress due to impaired ion uptake. As the authors used a much higher bacterial load (7.5×10^7 cfu/site in Fig 3A-B and 1.25×10^8 cfu/site in Fig 3C-D) compared to all other figures (approximately 2×10^6 cfu/site), is it possible that high load of lgt-mutant strain has impaired growth rate in the infected tissue, thus low bacteria burden observed due to loss of cell viability?

5. Control treatment of using Lpl1(-sp) + lgt-mutant SA113 strain should be included in in some experiments (Fig 5, 7C-7D).

6. In Fig 5C and 7G, additional time-point bacterial count at day 1 or 2 will be necessary to justify that Lpp and fibrin capsules protect bacteria from immune cells killing.

7. The authors claim that cyclophosphamide-treated mice lost weight and developed more severe diseases during the course of infection (Line 205 – 208), but no evidence were shown. The authors should include the data for proper interpretation.

Minor issues:

1. In Abstract, "Lpp-deficient *S. aureus* strains exhibited smaller lesion size and reduced bacterial loads than their parental strains; this altered phenotype was TLR2-independent.", but this is not consistent with the data, which suggest the skin lesion severity was TLR2-dependent at the early infection.

2. It would be great to include the FACS dot plots of immune cells isolated from auricular skin tissue following s.c. injection of PBS or Lpl1 in TLR2^{-/-} mice.
3. It would be interesting to add the role of IL-1R and MyD88 to the discussion.
4. Have the authors performed the bacterial cfu count in the supernatant of skin biopsy homogenates on day 10 post-infection with wild type or lgt-mutant SA113 strains (Fig 3C). As the authors showed the bacterial counts of Newman strain on day 10 in Fig 4C, it would be interesting to see if SA113 strain has similar effect to the Newman strain.
5. Have the authors accessed the levels of tissue factor and PAI-1 after s.c. injection of Lpl1(-sp)?

Gothenburg, January 4th, 2021

We would like to thank referees for the insightful examination of our manuscript. In order to strengthen our conclusions, eight independent *in vivo* experiments and several immunological assays were performed as reviewers advised including:

- 1) The bigger sample size of wild-type and TLR2^{-/-} mice for FACS analysis are now used (please see Results page 7 and figure 2).
- 2) To study whether adaptive immunity is involved in Lpp induced disease phenotype in skin infections, SCID and control mice were s.c. infected with Newman and Newman Δlgt strains. Our data suggest that adaptive immunity plays minor role regarding the Lpp induced phenotype in skin infections (please see Results page 11 and figure 6).
- 3) To study whether the complementation of Δlgt mutant restores the phenotype, complemented strain SA113 Δlgt (pRB474::*lgt*) was used. Our data demonstrated that complementation fully restored the phenotype that was abrogated in SA113 Δlgt . (Please see results page 8-9 and figure 3).
- 4) To understand whether lower bacterial doses change the Lpp induced effect in skin infection model, 20 times lower bacterial dose of both SA113 and SA113 Δlgt were used. Our results showed that SA113 induced more severe skin lesions and higher bacterial counts than SA113 Δlgt even with lower doses (Please see results page 8 and supplementary figure 2).
- 5) To study whether fibrinogen depletion promotes the bacterial metastasis, the bacterial counts from different organs (kidneys, spleen, liver, lungs, and blood) were performed. Our results suggest that fibrinogen depletion did not promote the bacteria dissemination to the other organs. (Please see results page 13 and supplementary table 1)
- 6) To study whether the chemokine release induced by Lpp in local skin is TLR2 dependent, the WT and TLR2^{-/-} mice were s.c. injected with purified Lpl1. Our data clearly showed that *in vivo* chemokine release induced by Lpp in local skin is TLR2 dependent (Please see results page 6 and figure 1I-K).
- 7) To verify whether lipid moiety of Lpp is of importance for the effect induced by Lpp in the skin infection caused by Lpp mixed with SA113 Δlgt , Lpl1(-sp) that lacks the lipid moiety was mixed with SA113 Δlgt instead and s.c. injected to the mice. Our data suggest that the lipid moiety of Lpp is crucial for the disease phenotype observed such as bigger skin lesions, higher bacterial loads, higher chemokine and MPO levels (Please see results page 11 and figure 5H-M).
- 8) To understand the effect of Lpp in skin infections at the early disease course, control substance or Lpl1(+sp) was mixed with live SA113 Δlgt strain and s.c. injected to the mice. Our results demonstrate that at the early timepoints when the fibrin capsules were not formed yet, the bacterial load in the local skins was similar in two groups (Please see results page 10 and supplementary figure 4).

Below please find our responses to all the queries, point by point.

Manuscript ID Ref.: **COMMSBIO-20-1679**- Title: “*Staphylococcus aureus* lipoproteins induce skin inflammation and promote abscess formation, shielding bacteria from immune killing”

Reviewers' comments:

Reviewer #1 (Remarks to the Author):

In “Staphylococcus aureus lipoproteins induce skin inflammation and promote abscess formation, shielding bacteria from immune killing”, Majd Mohammad and colleagues examine immune effects of lipoproteins of the human pathogen Staphylococcus aureus. They show that lipoproteins as proinflammatory, and bacteria deficient in them are less inflammatory. These observations are not entirely novel, but of interest and potential importance. However, key experiments are lacking adequate controls and do not fully support the conclusions. Specific notes follow:

Major comments

Altogether, significance appears to be usually reached based on animal number, and not necessary biological mechanism. Power functions need to be used to determine appropriate sample numbers which should be applied more consistently. In some experiments large numbers of mice are used to make fairly minor points (eg 32 mice in Fig 5C), while others are very underpowered (Figure 2 makes conclusions on the importance of TLR2 based on groups of 3 mice). Many appropriate comparisons appear that they would reach significance with adequate mouse numbers (for example, monocyte and macrophage number induced changes induced by Lpl1 in TLR2 mice, which does not agree with the proposed molecular mechanisms). In many paired datasets, the number of animals changes, suggesting samples are being omitted or other factors are confounding analysis. Its also not clear how many experiments are performed, when samples are combined, or are representative, or these differences decided.

: We are very sorry for confusing data presentation. In figure 5C, we performed three independent experiments, which exhibited similar results (2 of 3 with $p < 0.05$ and the third one was also close to significance), and data were thus pooled. Only samples from one representative experiment were used for chemokines and MPO assays. We have now added this information into the figure legend. The reason why we used 32 mice for this experiment is that we have recently discovered that intraarticular injection of mixture of Lpp with SA113 Δ lgt actually induced elimination of SA113 Δ lgt in joints (Mohammad et al, PloS Pathogens 2019). The results from s.c injection were completely opposite to i.a. injection experiments. To make sure the finding was true, we did 3 independent experiments with altogether 32 mice. We can of course use all data from one experiment if you prefer.

We fully agree with reviewer 1. Even through the differences were striking in FACS analyses, it is still necessary to increase the sample size to have statistical power. We have now repeated the FACS experiment with additional samples to verify our hypothesis, and as expected the new results are fully in line with the previous experiments and now merged and presented in figure 2. In our presented experiments, no samples are omitted but all samples are included.

Lpp deficient bacteria, expectedly, have decreased viability and growth, for reasons including deficiencies in iron acquisition. These mutants are more susceptible to killing by host immune effectors and are rapidly cleared – any mutant that so fundamentally impacts bacterial survival will decrease inflammation, but this is very non-specific and not a mechanism. This has not been sufficiently discussed or deconvolved as a contributor to the in

vivo observations. The attenuation of Lgt mutants also appears highly variable between strains and experiments. Prior literature is all consistent with these mutants being highly attenuated, which makes sense, but why this is seen in some experiments and not others is of concern. Potentially there are issues with mutants, or these are not fundamental mechanisms, but a phenomenology of certain strains that would at least require additional explanation.

: We agree. Now we have adjusted the discussion. The fibrin capsule formation is another mechanism above of the iron acquisition mechanism.

Line 147, it is not clear that the wound healing process was “faster”, as stated. The data are consistent with healing starting sooner, because the infection is resolved soon, since the bacteria are attenuated. Line 176; similar comment. The apparent role of lpp early in infection is likely due to attenuation and reduced bacterial burden; evidence is not consistent with roles of TLR2 signaling in this. The slopes of lesion size (figure 4, for example) suggest wound healing is not different, just starting from a different size.

: We have rephrased our conclusion that is more objective. We hope you agree with us.

Line 216. Also consistent with this model is that cyclophosphamide depletion of important immune cells reduces the killing of staph, even leading to a partial attenuation reversal of the lpp mutant strain.

Fibrin is a barrier to bacterial dissemination; therefore, ancrod depletion of fibrinogen is expected to promote staph dissemination. Figure 7 shows smaller lesions with ancrod treatment, with correspondingly fewer bacteria, but it is not clear that there just isn't fewer bacteria due to spread to other sites.

: Thank you for the constructive suggestion. It is obviously important to rule out this possibility. In order to address this question, we repeated the experiment and assessed the bacterial load in various organs, such as the blood, kidneys, liver, spleen, lungs, and also collected larger skin biopsies (double-sized biopsies than usual) of the mice treated with either PBS or Ancrod followed by subcutaneous injection of S. aureus Newman parental strain or Newman Δ lgt mutant strain. No CFU was found in the Ancrod treated mice (0/4), while 1 out of 4 mice in the PBS-control group displayed positive counts in the kidneys, liver, spleen and lungs, but not in blood. Our data suggest that depletion of fibrinogen does not promote S. aureus dissemination. These data are now included in supplementary table 1. For the CFU counts from the larger skin biopsies, the new results were very similar as the previous data. Please see figure below:

Minor comments

Minor edits for grammar needed throughout

Greater discussion on the number of Lpp proteins and their functions is needed

: The number of Lpp proteins and their functions are now provided in introduction.

No details on purity of Lpl1

: Lpl1 was prepared as previously described. We have now added the information about the purity of Lpl1 as reviewer suggested.

Line 167, discussion on the significance of agr+ would be useful

: We would like to thank review once again for insightful examination. The same comment was addressed by reviewer 3 and 4. Now we have added more discussion on it.

Are cytokines normalized by lesion size? Methods on sample collection could use more detail. Cytokines, for example, on Y-axis of graphs appear to be normalized to tissue mass. However, methods indicate skin was consistently collected by 8mm biopsy punch. This would be irrespective of lesion size, so small lesions will inherently have the cytokine numbers diluted out by normal tissue.

: We thank the referee for this thoughtful comment. Below are the reasons why we did not use the normalized cytokine levels.

Firstly, a few infection sites did not show any signs of skin lesion (lesion size= 0) and all healthy skins had the lesion size 0. This means we will lose some samples if we used the adjusted values (cytokine levels/lesion size).

If we would show our cytokine results calculated by 1 mm² tissue area, then it would be important to normalize the cytokine level for lesion size. We have not normalized the cytokines by lesion size since we show the total amount of cytokine measured in the whole tissue. The reason for this is that in the normal skin the cytokine level per 1 mm² is very low and it does not affect the final outcome.

An example of MCP-1:

Since all biopsies were taken with an 8 mm biopsy punch, the total area for all biopsies was 50.27 mm². We consider that the healthy mice without skin lesions reflect normal basal levels

of cytokines. The level of MCP-1 in the homogenized 8 mm biopsy tissue of 5 healthy mice ranged 18-54 pg/ml and the mean (and also median) level of MCP-1 was 35 pg/ml. In healthy skin, the MCP-1 level per 1 mm² skin tissue is therefore 0.7 pg/ml.

Considering for example a tissue sample with total MCP-1 level 2233 pg/ml and lesion size 24.63 mm² - the area of healthy skin would then be 25.64 mm² (total area 50.27mm² minus 24.63mm²) and the amount of MCP-1 cytokine coming from this area is 18.01 pg/ml (area of healthy skin 25.64 mm² multiplied by the median amount of MCP-1 in the healthy skin i.e. 0,7 pg/ml). This means that most of the MCP-1 amount comes from the lesion 2233-18=2215 pg/ml.

Considering the sample with total MCP-1 level 980 pg/ml and lesion size 4.9 mm², the area of healthy skin would be 45.36 mm² and the amount of cytokines from healthy area 31.8 pg/ml. The amount of MCP-1 from the lesion area would be 980-32=948 pg/ml.

Sample with MCP-1 level 217 pg/ml and lesion size 4.9mm²; the amount of MCP-1 from the lesion area would be 217-32=185 i.e. still much higher than the baseline levels of MCP-1 in the healthy skin.

Obviously, the majority of MCP-1 comes from the lesion and the relationship remains the same between total MCP-1 amount (indicated in our figures) and the amount calculated/released from skin lesion.

Histology lacks scale bars

: Thank you for the comment. This is now added in the figure legends of figure 1 and 7, respectively.

Reviewer #2 (Remarks to the Author):

This study shows that the lipid moiety of *S. aureus* lipoprotein (Lpp) contributes to the pathogenesis of skin infection by stimulating inflammation and abscess formation, using purified Lpp (Lp11) and *S. aureus* lacking lipidated Lpp (Δ lgt mutant) in mice. The authors also suggest that Lpp-induced abscess formation contributes to staphylococcal immune evasion given that fibrinogen depletion promoted WT *S. aureus* clearance but not the Δ lgt mutant. While interesting, these are not novel concepts (Schmaler et al. J Immunol June 1, 2009, 182 (11) 7110-7118, Cheng et al., FASEB J. 2009 Oct; 23(10): 3393–3404). I have the following comments:

1. The interpretation of data in the manuscript is often confusing or at times not quite accurate (see point 4 below please). The discussion section should be more concise.

: Now we have revised the discussion section. We hope the discussion is now acceptable.

2. The inconsistency in the role of TLR2 stimulation by Lpp upon the use of purified Lpp (Lp11) versus during bacterial infection was not clearly explained. While Lp11-induced skin lesion was dependent on TLR2, the Δ lgt mutant was attenuated in both the WT and TLR2^{-/-} mice. The latter observation could be due to the fact that the Δ lgt mutant has a growth defect under nutrient limitation compared to the parent strain as previously shown by Stoll et al. (Infect Immun 2005, 73(4):2411-2423) and the authors (Mohammad et al., Scientific Reports 2020, 10: 7936).

: We fully agree with you. The growth defect under nutrient limitation should not be underestimated. We have now modified the discussion section.

3. The authors show that during skin infection TLR2 does not control *S. aureus* infection (Fig. 4C, no increase in bacterial burden in TLR2-/-infected mice) but fail to mention and discuss this observation which corroborates that of Miller et al. (Immunity 2006, 24:79-91) but is in contrast to that of Hoebe et al. (Nature 2005, 433:523-527). The differing roles of TLR2 in skin infection have been suggested to be due to different bacterial strains used and different infectious doses administered in the skin. What doses of bacteria were given to the mice? The methods section only mention bacterial volumes.

: The doses of bacteria were described in the figure legends since the several different doses were used in the different experiments. Anyway, we have now discussed this issue and cited all the references.

4. (Page 10 lines 199-217) Paragraph title “The effect induced by Lpp expression is abrogated by leukocyte depletion”. This paragraph focuses on the attenuation of the Δ lgt mutant compared to the parent strain depending on the presence of leukocytes when in fact what this shows is that leukocytes are important to contain *S. aureus* skin infection.

: Thanks for the constructive comment. We apologize for the unclear paragraph title. Now we have rewritten it to make it clearer and more objective.

5. Line 206 - “cyclophosphamide-treated mice developed more severe disease”, can you please show the statistical significance in CFUs from cyclophosphamide-treated versus untreated mice (Fig. 6B)?

: The same comment was also addressed by other reviewers. This has now been presented in Fig. 6B, as the reviewer suggested.

6. Did you verify leukocyte depletion upon cyclophosphamide-treatment? (e.g. Flow data?)

: Yes, we verified the leukocyte depletion by measuring the level of peripheral mouse blood leukocytes in a cell counter. These data are now included in supplementary figure 7, showing successful depletion of leukocytes by cyclophosphamide.

7. Lines 207-208 – “The leukopenic mice also lost significantly more weight compared to ...infection”. No figure callout, where are the data?

: We apologize for this. These data are now included in supplementary figure 5.

Minor comments:

1. Line 49 “unique structure” of *S. aureus*, please rephrase

: Modified as suggested.

2. Line 65 *S. aureus* ‘contributes’, line 350 Lpp ‘plays’

: Modified as suggested.

3. Line 120 Lpp ‘induces’

: Modified as suggested.

4. Why was the immune cell analysis done in infected mouse auricles and not from the skin biopsies (injected mouse backs as per methods)?

: The skin biopsies from backs are extremely difficult to digest, prepare the single cells, and thereafter perform the FACS analysis. We have therefore utilized the well-established method of using mouse auricles. This system closely resembles the skin tissues and is much easier to work with in order to understand the immune response in vivo.

5. Line 161-162 – “SA113 Δ lgt mutant strain tended to induce lower levels of MPO in the skin tissue than SA113 parental strain (Fig. 3D). But there was no significant difference according to that figure?

: Correct, we have now clarified that in the result section. Please see line 194-196.

6. Lines 337-338 – “Bacterial MAMPs like LPS”, what is the relevance to *S. aureus*?

: It is a very good point and we fully agree. This part has now been removed.

7. Fig. 5G – MPO levels much lower compared to Fig. 3D, do these fluctuate that much?

: These are two independent experiments with different concentrations of bacteria. Higher MPO levels are associated with higher dose (1.25×10^8 cfu/site) of bacteria in figure 3D (updated to Fig 3G). In figure 5G only 2×10^6 cfu/site of bacteria were used and this set contains the mixture of purified LplI with the live bacteria. In summary, the fluctuation of MPO levels in those two experiments was due to different bacterial doses.

Reviewer #3 (Remarks to the Author):

Summary

In this study, authors showed Lipoprotein(Lpp) is necessary to recruit leukocytes and induce inflammation through TLR2 signaling. They also showed *S. aureus* with Lpp enhances skin abscess formation caused by imbalanced coagulation/fibrinolysis hemostasis in the local skin tissue by using Lpp deficient strain(Δ lgt). It is interesting idea that abscess formation is the bacterial evasion mechanism. But there is important question how does host immune function fight against Lpp deficient bacteria without innate immune response? Since Newman Δ lgt has slow growth rate in biological conditions (Mohammad, M. et al. The role of *Staphylococcus aureus* lipoproteins in hematogenous septic arthritis. Scientific Reports 10, 7936, doi:10.1038/s41598-020-537 64879-4 (2020)), it would be difficult to simply compare the number of bacteria after 3 or 10 days.

Also, authors use two different strains, SA113(Agr +) and Newman(Agr -), in different experiment. What is the impact from Agr system in this phenotype? It would be better to clarify in result or discussion.

: It is a very good comment (review 1 and 4 also commented on this). Now we have added the discussion in the manuscript.

Overall impression

The idea that bacteria take advantage of abscess formation to survive in host homeostasis, and their strategy is straight forward. But they need to address those questions.

Comments/Suggestion for authors

Major Queries

1, In figure 3 authors showed the SA113 Δ lgt strain cause smaller infectious lesion and less bacterial counts (Figure3C). It looks small difference and is nuclear the original number of injected SA113. Figure 3D shows SA113 Δ lgt has similar MPO level to WT strain. But in Figure 4B and D, Newman Δ lgt strain still cause smaller infectious lesion and less bacteria even in TLR2-/- mice. Furthermore, in figure 6, Cyclophosphamide treated leukocyte depletion mouse injected Newman parental strain didn't show higher larger skin lesion compare to PBS treated mice. Taken together it suggest Lpp is necessary to cause inflammation but bacterial survival or overload is due to the different growth speed of each

bacteria. In other words, the low number of bacteria is caused by slower growth of Δ lgt strain.

Could you address how does host fight against Lpp deficient bacteria without innate immune response? Is there any difference in adaptive immunity? Or is it possible to knock in Lpp in Newman or SA113 Δ lgt strain fo repeat experiment?

: We would like to thank reviewer very much for the constructive suggestion.

- 1) Now we have investigated the impact of adaptive immunity on the Lpp induced phenotype in skin infections using SCID mice who lack of both B and T cells. The Lpp induced phenotype was retained in SCID mice, suggesting adaptive immunity played minor role there. Please see figure 6C and 6D.*
- 2) Importantly, all phenotypes were restored when Lpp was knocked into SA113 Δ lgt strain. Please see figure 3H-M*

2, In figure 4, Authors use Newman(Agr -) strain instead of SA113(Agr +) for TLR2 Ko mice experiment. But there is no discussion about the impact of Agr sysmen. What is the impact of Agr system?

: Once again, very good comment (review 1 and 4 also commented on this). Now we have added the discussion about the impact of Agr system in the manuscript.

Minor Queries

1, The authors showed CFU in Figure 3D, 4C, 5C, 6B and 7G, but those are hard to see the difference even though there is statistically significant difference.

: The CFU data were presented in logarithmic scale Y axis, the significant differences were not so obvious in some cases. We have now tried to adjust the scales of Y axis in order to make the differences clearer. We hope the figures are now acceptable.

Reviewer #4 (Remarks to the Author):

Staphylococcus aureus lipoproteins (Lpp) are membrane anchored surface proteins that play important roles in host-bacterial interactions. How Lpp involved in S. aureus pathogenicity remain poorly studied. Mohammad and colleagues showed in their previous study that Lpp induces chronic destructive macroscopic arthritis (PMID: 31226163). In this study, Mohammad and colleagues further investigate the role of Lpp, in particular “lipoprotein-like” Lp11, in murine skin infection. They find that subcutaneous injection of Lp11 promote leukocytes infiltration and skin lesion, which are TLR2-dependent. Also, in comparison to wild type strain, lgt-mutant strain causes smaller skin lesion size and lower bacterial loads, which the latter is TLR2-independent. Finally, they find that the skin lesions and bacterial burden induced by Lpp can be abrogated by chemically-depletion of leukocyte or fibrinogen.

This study is potentially interesting, particularly lpp-induced fibrin capsule and abscess formation as a strategy to evade host immune attack, and will be of interest to a broad audience of infectious disease researchers and microbiologists. However, the mechanism underlying the abscess formation to protect S. aureus from immune cells is less convincing, as the conclusion was made largely based on experiments using purified Lp11 alone or co-injection with live bacteria. More rigorous examinations are required in this regard.

Below are comments/points that may help to improve the manuscript and hope the authors will find them useful.

: The reviewer 4 addressed several important questions and suggested the important experiments that significantly improved the quality of the manuscript. We highly appreciate the Reviewers input.

Major points:

1. The authors showed increased chemoattractant level after purified Lpl1 s.c. injection (Fig 1), thus promoting the infiltration of leukocytes to the injection site (Fig 2). However, the experiments heavily rely on purified Lpl1 treatment, an approach that may undermine other Staph virulence factors that also cause skin inflammation and leukocytes infiltration. It would be great if the authors could use lgt-mutant and complemented mutant lgt strains to validate the significant impact of Staph Lpp in chemoattractant release and leukocytes infiltration at the skin infection site.

: Very good comment that was also addressed by the other reviewers. Lgt mutant and complemented mutant were used in a new experiment. Skin lesion size, bacterial counts, the chemokine release, and MPO levels are now presented in fig 3H-M.

2. In general, TLR2 possess a protective function during *S. aureus* infection, which involves in neutrophils recruitment in response to *S. aureus*. The bacterial burden in the murine organs (Takeuchi et al. 2000, *J Immunol*, PMID: 11067888; Yimin et al. 2013, *Plos One*, PMID: 24058538) and skin (Miller et al. 2006 *Immunity*, PMID: 16413925) were higher in TLR2-deficient mice than wild type mice. In contrast, this study show a smaller skin lesion size and normal bacterial clearance that is similar to wild type mice (Fig 4). The authors should discuss this discrepancy.

: We appreciate this constructive comment. The other reviewers also addressed this question. Now we have discussed this discrepancy in the discussion section.

3. What are the levels of MIP-2, KC and MCP-1 of the skin biopsy homogenate in TLR2-/- mice after Lpl1 injection, as well as in WT vs TLR2-/- mice after WT and lgt-mutant strains infection? It would be interesting to see whether these chemoattractant releases are dependent on TLR2.

: Thank you for the constructive comment. In order to address these questions, we performed additional experiments, which generated three new subfigures as well as a supplementary figure. These data are now included in Fig. 11-K and in the supplementary figure 3. Our data clearly show that in case of purified Lpp, chemoattractant release was TLR2 dependent, whereas live bacterial setting was less conclusive but TLR-2 independent.

4. Despite the possibility of increased neutrophils and monocytes infiltration, higher bacterial burden was observed on day 3 after s.c. skin infection of both wild type SA113 and Newman strains compared to lgt-mutant strains (In Fig 3C and 4C). It is not convincing that lgt-mutant burden is significantly lower than wild type strain at day 3 as the difference is marginal (less than 2-fold). The authors also suggest that Staph Lpp provoked less bacterial clearance in wild type SA113 compared to lgt-mutant strain (Line 156 – 158). However, Stoll et al. 2005, *Infect Immun* (PMCID: PMC1087423) and Bubeck Wardenburg et al. 2006, *PNAS* (PMCID: PMC1564215) demonstrated that the growth of lgt-mutant strain was retarded under nutrient limitation and stress due to impaired ion uptake. As the authors used a much higher bacterial load (7.5×10^7 cfu/site in Fig 3A-B and 1.25×10^8 cfu/site in Fig 3C-D) compared to all other figures (approximately 2×10^6 cfu/site), is it possible that high load of lgt-mutant strain has impaired growth rate in the infected tissue, thus low bacteria burden observed due to loss of cell viability?

: Thank you for the constructive comment. The reason why the bacterial doses were much higher in the experiments performed in figure 3 is because SA113 is much less virulent compared to the Newman strain. In order to observe clinical signs of infection, a higher dose of SA113 was necessary.

Nevertheless, in order to address this question, we reduced the bacterial dose nearly 20 times and s.c. infected the mice. The results once again showed that the lesion size was more pronounced over time in the group subcutaneously infected with the SA113 parental strain. Also, the bacterial load in the skin of the mice tended to be different, but did not reach significance between the groups ($p=0.07$) on day 3 postinfection. We speculate this might be due to reduced bacterial dose that led to less inflammation and reduced abscess capsule formation, which somehow diminished bacterial evasion effect of fibrin capsule. These data are now included in supplementary figure 2.

5. Control treatment of using Lpl1(-sp) + lgt-mutant SA113 strain should be included in some experiments (Fig 5, 7C-7D).

: We fully agree. We have performed new experiments and the data are now included in figure 5H-M and figure 7E and 7F. As expected, no differences were observed when Lpl1 (-sp) was mixed with SA113 Δ lgt strain.

6. In Fig 5C and 7G, additional time-point bacterial count at day 1 or 2 will be necessary to justify that Lpp and fibrin capsules protect bacteria from immune cells killing.

: Very good point. We checked additional time point of bacterial count at day 1 when the fibrin capsules were not formed yet. No difference was found on day 1 after infection between WT and lgt mutant. The results are included in supplementary figure 4.

7. The authors claim that cyclophosphamide-treated mice lost weight and developed more severe diseases during the course of infection (Line 205 – 208), but no evidence were shown. The authors should include the data for proper interpretation.

: The same comment was addressed by other reviewers. The data are now presented in supplementary figure 5.

Minor issues:

1. In Abstract, “Lpp-deficient *S. aureus* strains exhibited smaller lesion size and reduced bacterial loads than their parental strains; this altered phenotype was TLR2-independent.”, but this is not consistent with the data, which suggest the skin lesion severity was TLR2-dependent at the early infection.

: We fully agree. The abstract is now modified.

2. It would be great to include the FACS dot plots of immune cells isolated from auricular skin tissue following s.c. injection of PBS or Lpl1 in TLR2-/- mice.

: Modified as reviewer suggested. These data are now included in supplementary figure 1.

3. It would be interesting to add the role of IL-1R and MyD88 to the discussion.

: The role of IL-1R and MyD88 are now added to the discussion.

4. Have the authors performed the bacterial cfu count in the supernatant of skin biopsy homogenates on day 10 post-infection with wild type or lgt-mutant SA113 strains (Fig 3C). As the authors showed the bacterial counts of Newman strain on day 10 in Fig 4C, it would be interesting to see if SA113 strain has similar effect to the Newman strain.

: We have never performed the cfu count on day 10 post infection, as SA113 is much less virulent than Newman strain and some of infection sites were already healed on day 10.

5. Have the authors accessed the levels of tissue factor and PAI-1 after s.c. injection of Lpl1(-sp)?

: The levels of TF and PAI-1 after injection of Lpl1(-sp) are now presented in figure 7A and 7B.

We hope that our improved manuscript will be now acceptable for publication in *Communications Biology*.

Yours sincerely,
Majd Mohammad, PhD

REVIEWERS' COMMENTS:

Reviewer #1 (Remarks to the Author):

The authors have addressed all points and I have no additional major concerns.

Reviewer #3 (Remarks to the Author):

Authors are properly addressed to all the questions from reviewers. Especially the skin infection model with SCID mice supports the importance of innate immune reaction.